# Much Ado About Noising: Dispelling the Myths of Generative Robotic Control

**Chaoyi Pan**[1], **Giri Anantharaman**[1], **Nai-Chieh Huang**[1], **Claire Jin**[1], **Daniel Pfrommer**[2],
Chenyang Yuan[3], Frank Permenter[3], Guannan Qu[1,‡], Nicholas Boffi[1,‡], Guanya Shi[1,‡],
Max Simchowitz[1,‡]

[1]Carnegie Mellon University, [2]Massachusetts Institute of Technology, [3]Toyota Research Institute
[†]Equal advising.
{chaoyip, giria, naichieh, clairej}@andrew.cmu.edu
{gqu, nboffi, gshi, msimchow}@andrew.cmu.edu
dpfrom@mit.edu, {chenyang.yuan, frank.permenter}@tri.global

## Abstract

Generative models, like flows and diffusions, have recently emerged as popular and efficacious policy parameterizations in robotics. There has been much speculation as to the factors underlying their successes, ranging from capturing multi-modal action distribution to expressing more complex behaviors. In this work, we perform a comprehensive evaluation of popular generative control policies (GCPs) on common behavior cloning (BC) benchmarks. We find that GCPs *do not* owe their success to their ability to capture multi-modality or to express more complex observation-to-action mappings. Instead, we find that their advantage stems from *iterative computation*, as long as intermediate steps are supervised during training and this supervision is paired with a suitable level of *stochasticity*. As a validation of our findings, we show that a minimal iterative policy (**MIP**), a lightweight two-step regression-based policy, essentially matches the performance of flow GCPs, and often outperforms distilled shortcut models. Our results suggest that the distribution-fitting component of GCPs is less salient than commonly believed, and point toward new design spaces focusing solely on control performance. Code and supplementary materials are available at our website.

## 1 Introduction

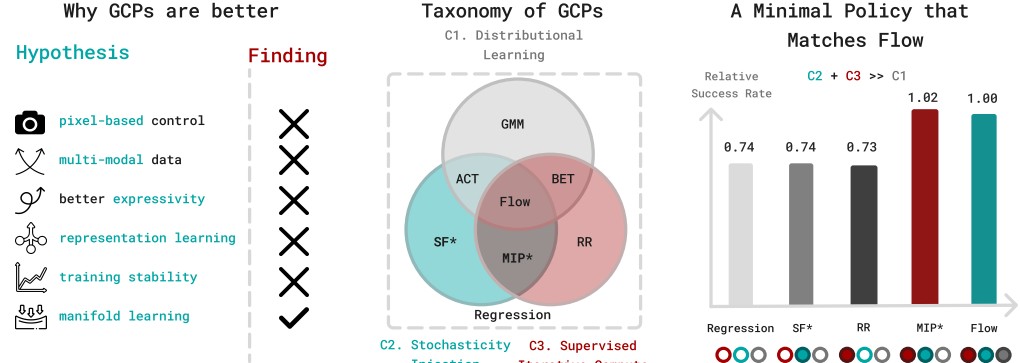

**Figure 1:** *Left:* After careful ablation on each component over 28 common behavior cloning benchmarks with diverse input modalities (state, pixel, point cloud and language), architectures (with both raw and pre-trained models, like $\pi_0$) and tasks (standard single-task benchmarks and multi-task benchmark like LIBERO), we refute a number of popularly held misconceptions about why **generative control policies** (GCPs) outperform regression policies (RCP) on these tasks. *Center:* We identify that the most important factor contributing to GCP success is a combination of *stochastic injection* (C2) and *supervised iterative computation* (C3). Surprisingly, distribution learning (C1) is the least important factor, due to the absence of learned multi-modality (Section 3.2). *Right:* The average relative success rate to flow of 7 most challenging tasks. We propose a simple two-step **minimal iterative policy** (**MIP**) whose performance matches that of flow-based GCPs.

Long-horizon, dexterous manipulation tasks such as furniture assembly, food preparation, and manufacturing have been a holy grail in robotics. Recent large robot action models (Team et al., 2025; Black et al., 2024; Kim et al., 2024) have made substantial breakthroughs towards these goals by imitating expert demonstrations of diverse qualities. We provide a more comprehensive review of related work in Section A, but highlight here a key trend: while supervised learning from demonstration, also known as *behavior cloning* (BC), has been applied across domains for decades (Pomerleau, 1988), its recent success in robotic manipulation has coincided with the adoption of what we term **generative control policies** (GCPs): robotic control policies that use generative modeling architectures, such as diffusion models, flow models, and autoregressive transformers, as parameterizations of the mapping from observation to action. Given the seemingly transformative nature of GCPs for robot learning, there has been much speculation about the origin of their superior performance relative to policies trained with a regression loss, henceforth **regression control policies** (RCPs). GCPs, by modeling conditional distributions over actions, are uniquely suited to the multi-task pretraining paradigm popular in today's large robotic models. However, a number of hypotheses regarding the superiority of GCPs pertain even in the *single task* setting (Chi et al., 2023; Reuss et al., 2023):

H1. Better performance on pixel-based control
H2. Capturing multi-modality in the training data
H3. Greater expressivity due to iterative computation of the observation-to-action mapping
H4. Representation learning due to stochastic data augmentation
H5. Improved training stability and scalability

**The gap between generative modeling and generative control.** The objective for generative modeling in text and image domains is fundamentally different from the goal in a control task. In the former, one aims to generate high-quality and *diverse* samples from the original data distribution. In the latter, it suffices to select *any* action that leads to better downstream performance. Whereas much of the generative modeling literature has focused on the distribution of the *generated variable* (Lee et al., 2023), we aim to understand if it is necessary to reproduce the expert data distribution— for example by capturing any multi-modality—to attain strong control performance. If not, is most salient to capture about the *conditioning relationship* mapping $o \rightarrow a$?

**Contributions.** This paper adopts careful experimental methodology to rigorously test the key design components (Section 4) that contribute to the observed success of GCPs, and to account for the key mechanisms by which they contribute to improved performance in behavior cloning (Section 5). We restrict our study to flow-based GCPs, given their popularity and adoption in industry (Black et al., 2024; Physical Intelligence et al., 2025; NVIDIA et al., 2025).

We begin by first identifying which factors *do not* contribute to the advantage of GCPs over RCPs.

**Contribution 1** (**Neither multi-modality nor policy expressivity account for GCPs' success**, Section 3). Through careful benchmarking, we show that RCPs with appropriate architectures are highly competitive on both state- and image-based (H1) robot learning benchmarks as well as vision-language-action (VLA) model finetuning (Section 3.1). Performance gaps only arise on certain tasks requiring high precision. However, we show that neither multi-modality (H2, Section 3.2) nor the ability to express more complex functions via multiple integration steps (H3, Section 3.3) satisfactorily accounts for this phenomenon. In fact, GCPs do not even provide greater trajectory diversity compared to RCPs (Section I).

Essential to this finding is controlling for architecture: to our knowledge, we are the first work to carefully benchmark expressive architectures popularized for Diffusion (Chi et al., 2023; Dasari et al., 2024) as regression policies. To determine what contributes to GCPs performance on these high-precision tasks (beyond architectural optimization), we parse the design space of generative control policies into three components, depicted in Figure 1 (left).

**Contribution 2** (**Exposing the design space of GCPs**, Section 4). We introduce a novel taxonomy that parses the three essential design components of GCPs:

C1. *Distributional Learning*: matching a conditional distribution of actions given observations.
C2. *Stochasticity Injection*: injecting noise during training to improve the learning dynamics.
C3. *Supervised Iterative Computation*: generating output with multiple steps, each of which receives supervision during training.

With this taxonomy in hand, Section 4.1 introduces a family of algorithms, each of which lies along a spectrum between GCPs and RCPs by exhibiting different combinations of the above components. While we find that neither C2 nor C3 in isolation improve over regression, we find their combination yields a policy whose performance is competitive with flow, leading to our next contribution.

**Contribution 3** (**MIP: the power of C2+C3**, Sections 4.1 and 4.2). As an algorithmic ablation that only combines C2+C3, we devise a *minimal iterative policy* (**MIP**), which invokes only two iterations, one-step of stochasticity during training, and deterministic inference. Despite its simplicity, **MIP** essentially matches the performance of flow-based GCPs across state-, pixel- and 3D point-cloud-based BC tasks, exposing that the combination of C2+C3 is responsible for the observed success of GCPs. In addition, we find that **MIP often outperforms shortcut/few-step policies** (Section D.8). This confirms our findings that distributional learning (which few-step policies, but not **MIP**, achieve) is not needed in robotic control.

As described in Section D.8, **MIP** is substantively distinct from flow-map-based models (Boffi et al., 2025a;b), including consistency models (Song et al., 2023; Kim et al., 2023) and their extensions (Geng et al., 2025; Frans et al., 2024), in that the latter do satisfy C1, and require training over a continuum of noise levels.

**Contribution 4** (**Attributing the benefits of C2+C3**, Section 5). We identify that a property we term *manifold adherence* captures the inductive bias of GCPs and **MIP** relative to RCPs, even in the absence of lower validation loss. We explain how this property is a useful proxy for closed-loop performance in control tasks. Finally, we expose how C3, through iterative computation, encourages manifold adherence, but only if stochasticity during training (C2) is present to mitigate compounding errors across iteration steps (as described in Section 5.2).

Manifold adherence in Section 5.1 measures the generated action's plausibility given out of distribution observations, where only off-manifold component is evaluated rather than the distance to the neighbors (Pari et al., 2021). Note that manifold adherence reflects a favorable inductive bias during learning, rather than brute expressivity of more complex behavior (H3). Moreover, C2 provides more of a supporting role to C3, rather than enhancing data-augmentation in its own right (H4). In addition, we find that C2+C3 also enhance scaling behavior (H5), likely due to better model utilization through decoupling across iterations. Finally, we identify that the subtle interplay between architecture choice, policy parameterization and task can affect performance by an even greater magnitude than the choice of policy parametrization (Section 5.3).

**Takeaway.** In robotic applications, our findings suggest that the distributional formulation of GCPs — sampling from a *distribution* of actions given observations — is the least important facet that contributes to their success. Rather, our work highlights that C2+C3 offer an exciting and under-explored sandbox for future algorithm design in continuous control and beyond.

## 2 PRELIMINARIES

We consider a continuous control setting with observations $o \in O$ and actions $a \in A$ where $O$ is the observation space and $A$ is the action space. We learn a policy $\pi : O \to \Delta(A)$ from observations to (distributions over) actions to maximize the probability of success $J(\pi)$ on a given task, which we refer to as "performance." We consider the performance of policies learned via BC—that is, supervised learning from a distribution of (observation, actions pairs) drawn from a training distribution $p_{\text{train}}$. In applications, the actions $a$ are often a short-open loop sequence of actions, or *action-chunks*, which have been shown to work more effectively for complex tasks with end-effector position commands (Zhao et al., 2023b). See Section A for an unabridged related work.

**Regression Control Policies (RCPs).** A historically common policy choice for BC is regression control policies (RCPs) (Pomerleau, 1988; Bain & Sammut, 1995; Ross et al., 2011; Osa et al., 2018), given by a deterministic map $\pi : O \to A$. In applications, it is parameterized by a neural network $\pi_\theta$ and trained so as to minimize the $L_2$-loss on training data:

$$\pi_\theta \approx \arg\min_\theta \mathbb{E}\|\pi_\theta(o) - a\|^2, \quad (o, a) \sim p_{\text{train}}. \tag{2.1}$$

**Generative Control Policies (GCPs).** Generative control policies (GCPs) parameterize a *distribution* of actions $a$ given an observation $o$. This is often accomplished in practice by representing the

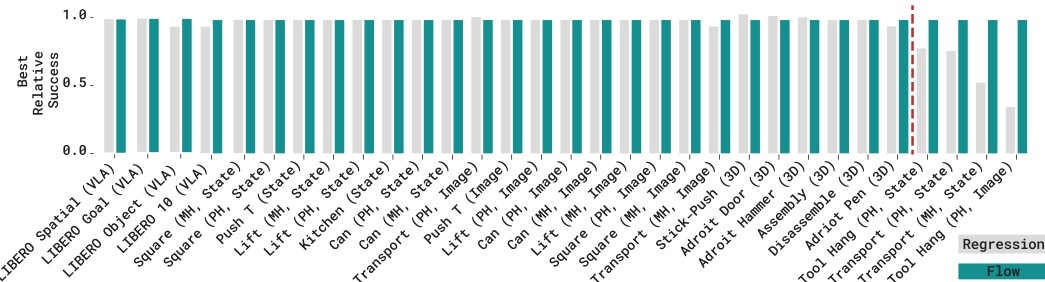

**Figure 2: Relative performance of RCPs compared to GCPs across common benchmarks.** For single-task benchmarks, we implement `Chi-Transformer`, `Sudeep-DiT` and `Chi-UNet`. For each architecture, we average performance of the best training checkpoint across 3 seeds. For multi-task benchmarks, we use $\pi_0$ as base policy and finetune it on full `LIBERO` benchmark (130 tasks). We then report the performance of the best-performing architecture, chosen individually for both RCPs and GCPs. For Flow, we always do 9 step Euler integrations, where its performance plateaued. For readability, RCPs success rates are plotted relative to flow, with flow normalized to performance of 1 per task. Tasks are grouped by observation modality, and ordered by relative RCPs performance. Red dashed line indicates threshold at which RCP attains $< 95\%$ success of GCPs. *Note that RP and* **Flow** *perform comparably on most Image, 3D-based and VLA-based multi-task benchmarks.*

policy $\pi_\theta$ with a generative model such as a diffusion (Chi et al., 2023), flow (Zhang et al., 2024), or tokenized autoregressive transformer (Shafiullah et al., 2022). Given their popularity, we focus on flow-based GCPs (flow-GCPs). A flow-GCP learns a conditional flow field (Lipman et al., 2023; Chisari et al., 2024; Nguyen et al., 2025; Albergo & Vanden-Eijnden, 2022; Heitz et al., 2023; Liu et al., 2022) $b : [0, 1] \times A \times O \to A$ by minimizing the objective

$$b_\theta \approx \arg\min_\theta \mathbb{E}\|b_t(I_t \mid o) - \dot{I}_t\|^2, \quad t \sim \text{Unif}([0, 1]), \quad z \sim \text{N}(0, \mathbf{I}), \quad (2.2)$$

where again $(o, a) \sim p_{\text{train}}$, $I_t = ta + (1 - t)z$ is the stochastic interpolant between the training action $a$ and noise variable $z$, and where $\dot{I}_t = a - z$ is the time derivative of $I_t$. We note that this is a special case of the stochastic interpolant framework (Albergo & Vanden-Eijnden, 2022; Albergo et al., 2023; 2024), which permits a larger menu of design decisions. A flow model then predicts an action by integrating a flow. In the limit of infinite discretization steps, this amounts to sampling $a \sim \pi_\theta(\cdot \mid o)$ by sampling $z \sim \text{N}(0, \mathbf{I})$, and then setting $a = a_1$, where $\{a_t\}_{t \in [0,1]}$ solves the ODE:

$$\frac{\mathrm{d}}{\mathrm{d}t} a_t = b_t(a_t \mid o) \qquad \text{with initial condition} \qquad a_0 = z. \quad (2.3)$$

In practical implementation, sampling is conducted via discretized Euler integration (see Section M.2 for details). This yields a policy $a = \pi_\theta(z, o)$ which is a deterministic function of the initial noise $z$ and the observation $o$. All experiments, unless otherwise stated, perform 9 integration steps. We reiterate that other GCPs, e.g. based on diffusion models and autoregressive transformers, have been studied elsewhere. We choose to focus on flow models due to their state-of-the-art performance (Chi et al., 2023; Chisari et al., 2024; Zhang et al., 2024) and deployment in industry (Black et al., 2024; Physical Intelligence et al., 2025; NVIDIA et al., 2025).

**Multi-Modality in Robot Learning.** Past work has conjectured that for salient robotic control tasks, $p_{\text{train}}(a \mid o)$ exhibit *multi-modality*, i.e. the conditional distribution of $a$ given $o$ has multiple modes (Shafiullah et al., 2022; Zhao et al., 2023b; Florence et al., 2022). This motivated the earliest use of GCPs (Chi et al., 2023) (H2). Section 3.2 calls into question the extent to which GCPs do in fact learn multi-modal distributions of $a \mid o$ on popular benchmarks.

## 3 MULTI-MODALITY AND EXPRESSIVITY DO NOT EXPLAIN GCPS' PERFORMANCE

This section demonstrates that neither advantages on pixel-based control (H1), nor multi-modality (H2), nor improved expressivity (H3) fully account for the GCPs performance relative to RCPs. Instead, our analysis indicates that the advantage of GCPs is **largely due to architectural innovations** found in GCPs—specifically, the adoption of powerful models like Transformers and UNets, along with the use of action chunking techniques. Sections H and I addresses other hypotheses, such as $k$-nearest neighbor approximation and the behavior diversity.

### 3.1 WHEN CONTROLLED FOR ARCHITECTURE, GCPs ONLY OUTPERFORM ON FEW TASKS

We first isolate the tasks in which GCPs exhibit stronger performance by comparing across 28 popular BC benchmarks including multi-task benchmarks like LIBERO (detailed in Section D.1), encompassing diverse data quality, modalities (**state**, **point clouds**, **image** and **language**), and domains (e.g., MetaWorld, Robomimic, Adroit, D4RL, Meta-World, LIBERO). Crucially, we implement RCPs using the **exact same architectures** as their corresponding flow models by simply setting the noise level and initial noise to zero: $z = 0, t = 0$, and study three widely-used architectures (Chi-Transformer, Sudeep-DiT, Chi-UNet as well as pre-trained VLA models like $\pi_0$ (Black et al., 2024); detailed in Section D.2). This architectural alignment enables RCPs to benefit from the sophisticated network designs typically reserved for GCPs, ensuring a fair comparison.

Under controlled comparison, we find GCPs and RCPs achieve parity across the vast majority of BC benchmarks. Performance gaps emerge only on a small subset of tasks requiring **high precision** (e.g. precise insertion tasks). We report best-case results in Fig. 2 and comprehensive ablations (including worst-case architectures and loss variants) in Section D.4.

Our evaluation yields three key insights: (a)**Rare Benefit of GCPs:** GCPs outperform RCPs by $> 5\%$ on only a handful of tasks. (b)**Modality Independence:** Contrary to popular belief, observation modality does *not* correlate with GCP advantage. (c)**Architectural Dominance:** Architecture choice dictates performance far more than the generative vs. regression distinction.

We posit that the perceived superiority of GCPs in prior work was confounded by architectural asymmetry. To our knowledge, this is the first study to benchmark Sudeep-DiT, Chi-UNet, and $\pi_0$ backbones as regression policies. In Section 5.3, we demonstrate that when equipped with these modern backbones—or even tuned MLP baselines—RCPs are highly competitive. Furthermore, we find that hyperparameters such as **action-chunking** horizon (Zhao et al., 2023a; Chi et al., 2023; Zhang et al., 2025) exert a greater influence on success rate than the choice of objective function (Section H.1). **Design decisions like architecture and action-chunking have a significant and consistent impact on control performance. In contrast, the choice between GCPs and RCPs is largely negligible outside of high-precision regimes.**

### 3.2 GCPs' PERFORMANCE DOES NOT ARISE FROM MULTI-MODALITY

Earlier literature suggested that capturing multi-modality, as defined in Section 2, was precisely the root of the observed performance benefits of GCPs (Chi et al., 2023; Reuss et al., 2023). However, examining Fig. 2, we see that many tasks which have been understood to be multimodal (e.g., Push-T) do not show substantial performance gaps between RCPs and GCPs. On the other hand, RCPs and GCPs differ only on tasks that demand high precision (e.g. Tool-Hang, Transport). In this section, we provide additional evidence that **multimodality is not the main factor responsible for witnessed performance advantages of GCPs**.

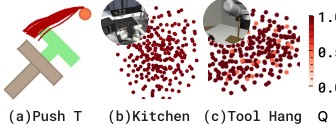

(a)Push T   (b)Kitchen   (c)Tool Hang   Q

| Task | $z \equiv 0$ | $N(0, \mathbf{I})$ | Mean $z$ |
|------|------|------|------|
| Push-T | 0.97 | 0.97 | 0.95 |
| Kitchen | 0.99 | 0.99 | 0.97 |
| Tool-Hang | 0.78 | 0.80 | 0.76 |

| Dataset | Flow | Reg. |
|---------|------|------|
| Original | 0.78 | 0.58 |
| Deterministic | 0.72 | 0.64 |

**Figure 3: A. Visualized action distribution with Q values.** Distinct modes are **not** observed in planned actions even at symmetric and ambiguous states. (Kitchen and Tool-Hang, t-SNE visualization.) In Push-T, we all trajectories goes to one side. For the rest, there is no clear clustering of actions or Q.

**Table 1: B. Performance comparison of different sampling strategies.** We compare sampling $z = 0$, $z \sim N(0, \mathbf{I})$, and mean over 64 $z^{(i)} \sim N(0, \mathbf{I})$. Different sampling strategies show minor performance difference, indicating absence of distinct action modes.

**Table 2: C. GCPs outperforms RCPs with deterministic experts.** Policy average success rate over 3 architectures, 3 seeds and 3 architectures given different dataset: one from original human demonstration and another collected by rolling out a flow policy in deterministic mode starting from zero noise.

**Evidence A: GCPs exhibit unstructured action distributions.** For fixed observations, we draw multiple action samples by denoising from different initial latents and visualize the resulting action set with their Q values $Q(a, o)$. We deliberately choose *symmetry-critical* or *high-ambiguity* states to *maximize* potential multi-modality: (a) Push-T at the symmetry axis of the T-shape, where taking

the left or right path is equivalent, (b) `Kitchen` from an initial state with multiple first-subtask choices, and (c) `Tool-Hang` at the insertion pre-contact pose where human demonstrators pause for varying durations. In (a-c) we observe *single* clusters rather than distinct modes (high-dimensional actions visualized with t-SNE); see Fig. 3. Moreover, adherence to action cluster means do not correlate with performance: We color-code actions by Q-value, i.e. Monte-Carlo-estimated rewards-to-go (Section F.1). Highest returns are distributed evenly across samples.

**Evidence B: Taking mean actions does not meaningfully degrade GCPs' performance.** We evaluate flow policy's performance with three sampling strategies: zero noise $a = \pi(z = 0, o)$, stochastic sampling $a = \pi(z, o), z \sim N(0, I)$, and *mean action* $a = \mathbb{E}_{z \sim N(0,I)}[\pi(z, o)]$ (via Monte Carlo approximation). If the learned distribution were multi-modal, or if their distributions lied on a manifold whose *curvature* was crucial to task success, the conditional mean would *collapse* modes and severely degrade performance. However, Table 1 shows that replacing stochastic sampling with the mean action only slightly affects performance, indicating absence of distinct action modes.

**Evidence C: GCPs outperform RCPs on certain tasks even with deterministic experts.** To fully remove any residual multi-modality, we recollect the dataset with trained flow policy evaluated in deterministic mode ($z = 0$) detailed in Section F.2. The new dataset is fully deterministic because action labels are provided by a deterministic policy evaluated in a deterministic environment. While the gap in performance between GCPs and RCPs shrinks somewhat, we still find that GCPs still outperforms RCPs, as in Table 2, suggesting that capturing some "hidden" stochasticity or multi-modality in the data does not suffice to explain the gap between the two. **Collectively, (A)–(C) indicate that the commonly cited explanation—"GCPs win because demonstrations are multi-modal"—does not hold for most studied behavior cloning benchmarks.**

**Multi-modality and data coverage.** The absence of observed multimodality is likely attributable to the large observation dimension of tasks relative to total number of demonstrations. That is, we rarely see two "conflicting" actions for nearby observation vectors (note: to grid a space of dimension $d$ requires $2^d$ points). Some degree of "hidden" multi-modality may still be present, as indicated by the slight narrowing of the performance gap in Table 2. Still, our central claim is that multi-modality is not *sufficient* to explain the full difference in performance.

### 3.3 LIMITATIONS OF THE EXPRESSIVITY OF GCPs IN THE ABSENCE OF MULTIMODALITY

An alternative to learning explicit multimodality is to represent rapid transition between actions as the observation changes. This is depicted in Figure 6, where data that appears multi-modal can be fit with a policy that has a high Lipschitz constant, i.e. in which $\nabla_o \pi(a \mid o)$ is large. This reflects a broader principle in control that we need only capture the mapping from observation to a single effective action, rather than reproduce the distribution over all possible actions.

One may still conjecture that GCPs more easily higher-Lipschitz policies by leveraging iterative computation, as compared to RCPs. This is because deeper networks can express larger-Lipschitz functions more easily (Telgarsky, 2016), and many have equated the multi-step computation in flow-based generative models to depth (Chen et al., 2018). Step-by-step generation is known to drastically increase expressivity in other domains as well, such as autoregressive language models (Li et al., 2024),

However, flow-based generative models use their multi-step computation to express complex distributions over the *generated variable* (Ho et al., 2020; Song et al., 2021a; Zhang & Chen, 2022; Nichol & Dhariwal, 2021). It is less clear if the iteration computation assists with represent complex *observation-to-action* mappings. Thus, we ask: **Does the iterative computation in GCPs aid in learning more complex observation-to-action mappings, even if the learned action distributions for a fixed observation are themselves are relatively simple (i.e. unimodal)?**

We now provide evidence that suggests "**no**." We show that in the absence of multi-modality (as shown in Section 3.2), GCPs cannot express more complex mappings from the conditioning variable $o$ to the generated variable $a$ than RCPs can. We begin by considering a ground-truth conditional flow field $b_t^\star(o \mid a)$. Let $\pi_\theta^\star(z, o)$ represent the exactly integrated $b^\star$ from initial noise $z$ to generated variable $a$. Given the absence of multi-modality (Section 3.2), we assume that the distribution of $a \mid o$ is $\kappa$-log-concave (Section J), satisfied by many classical unimodal distributions. We prove

that the Lipschitz constant of $\pi_\theta^\star(z, o)$ with respect to $o$, a measure of the expressivity of the $o \to a$ mapping, is bounded by that of $b_t^\star$:

**Theorem 1** (Informal). *Let $\| \cdot \|$ denote either the matrix operator or Frobenius norm, and suppose that the distribution of $a \mid o$ is $\kappa$-log-concave. Moreover, suppose that the flow field $b_t^\star(a \mid o)$ is $L$-Lipschitz: $\|\nabla_o b_t^\star(a \mid o)\| \leq L$. Then, with infinite integration steps, $\|\nabla_o \pi_\theta^\star(z, o)\| \leq L \cdot \sqrt{1 + \kappa^{-1}}$.*

See Section J for a formal statement and proof. A classical example of a log concave distribution is $a \mid o \sim \mathrm{N}(\mu(o), \frac{1}{\kappa})$; as long as the variance $1/\kappa$ is bounded *above* (even in the limit of a Dirac), there is at most a constant-multiplicative factor increase in the Lipschitz constant. When training a flow, $b_t^\star(a \mid o)$ is approximated by the neural network. Thus, in the prototypical unimodal example of $\kappa$-log-concave distributions, GCPs are not arbitrarily more expressive than RCPs. In fewer words: **more integration steps (i.e. more iterative computation), even infinitely many, need not enable greater expressivity of high Lipschitz $o \to a$ mappings**.

To verify our theoretical prediction, we quantify learned policies' Lipchitz constants with a zeroth-order proxy: starting from dataset states $s_t$ with observation $o_t$, we inject small Gaussian perturbations in the executed action to reach a *feasible* nearby state $s_{t+1}^{(i)}$ with observation $o_{t+1}^{(i)}$, then measure input–output sensitivity

| Method | Push-T | | Kitchen | Tool-Hang | |
|---|---|---|---|---|---|
| | State | Image | State | State | Image |
| **Regression** | 0.90 | 0.55 | 14.07 | 1.71 | 1.65 |
| **Flow** | 0.45 | 0.20 | 12.43 | 1.41 | 1.37 |

**Table 3: Policy Lipschitz constant comparison.** Lipschitz constant is averaged over 100 states.

via finite differences of the policy around the perturbed states (full algorithm and per-architecture results in Section E). This construction (i) avoids reliance on noisy higher-order gradients in complex architectures, and (ii) keeps evaluations on feasible observation to prevent conflating expressivity with model error on dynamically infeasible states. As predicted by our theory, GCPs are not strictly more expressive than RCPs as shown in Table 3. On the contrary, RCPs show increased Lipschitz constants off the manifold of training data, ruling out the assumption that GCPs win due to expressing policies with greater sensitivity to the input variable. We note that our methodology, which perturbs actions rather than states, is compatible pixel observations. To summarize: **In the absence of multimodality, GCPs do not enjoy an advantage over RCPs in expressing high Lipschitz behavior, such as rapid transitions between action modes.**

## 4 MINIMAL ITERATIVE POLICY (**MIP**): ISOLATING THE SOURCE OF GCPS' SUCCESS

In this section, we introduce a number of intermediates between RCPs and GCPs that isolate which design decisions contribute to the latter's superior performance. This leads to a Minimal Iterative Policy (**MIP**), which matches GCPs performance, thereby identifying the source of GCPs' success.

We begin with a taxonomy of the three key algorithmic components (Figure 1) present in GCPs. Section 4.1 below proposes algorithmic variants which ablate these components. We find that minimal iterative policy (**MIP**, Components 2 and 3) is the reduced variant which matches the performance of flow (Section 4.2), whereas other variants match or perform worse than regression.

**Component 1**. **Distributional learning** denotes training a model to fit a conditional distribution $a \sim \pi_\theta(o)$ of actions given observations, as opposed to deterministic predictions (i.e., $a = \pi_\theta(o)$). [1]

**Component 2**. **Stochasticity injection** denotes the injection of additional stochastic inputs into the neural network during training time (e.g., the variable $z$ in Eq. (2.2)).

**Component 3**. **Supervised Iterative Computation (SIC)** denotes the iterative refinement of predictions by feeding the previous outputs into the same network again during inference, and providing *supervision signals* at every step of the generation procedure at training time. For example, in flow GCPs, we integrate a supervised flow field $b_t(a_t \mid o)$ over time to get the final action $a$, and that $b_t$ receives an independent supervisory signal for each $t$ at training time (Eq. (2.2)).

## 4.1 MIP: A MINIMAL INTERMEDIATE BETWEEN RCPs AND GCPs

We introduce a range of policies which lie along the spectrum between RCP and flow-based GCPs via varying combinations of Components 2 and 3, culminating in the Minimal Iterative Policy (**MIP**). These policies do not satisfy Component 1, because Sections 3.2 and I suggests that this is not needed. In particular, we consider networks $\pi_\theta(o, I_t, t)$ that predict *actions*, not velocities, and given observations $o$, time indices $t$, and interpolants $I_t$ corresponding to noising actions. We state all networks below of $L_2$ minimization, but our findings remain consistent when minimizing $L_1$ error instead (Section H.2).

**Two-Step Denoising.** As a simplification of flow-based GCPs, we consider a **two-step denoising** (**TSD**) policy. As discussed in Section D.8, this parametrization is superficially similar to, but substantively different than, popular flow-map/consistency/shortcut models (Boffi et al., 2025b). **TSD** performs two steps of denoising, one from zero, and a second from a fixed index $t_\star = .9$:

$$\pi_\theta^{\texttt{TSD}} \approx \arg\min_\theta \mathbb{E}\left[\|(\pi_\theta(o, I_0 = z, t = 0) - (t_\star)^{-1} I_{t_\star})\|^2 + \|(\pi_\theta(o, I_{t_\star}, t_\star) - a)\|^2\right]. \quad (4.1)$$

where $(o, a) \sim p_{\text{train}}, z \sim \mathrm{N}(0, \mathbf{I})$, and $I_t = ta + (1 - t)z$ is the same interpolant used in flow models, and where $t_\star = .9$ is fixed. The normalization by $t_\star$ in Eq. (4.1) comes from the identity $t_\star a = \mathbb{E}_z[I_{t_\star}]$. We then sample $\hat{a}_0^{\texttt{TSD}} \leftarrow \pi_\theta(o, z, 0)$ and $\hat{a}^{\texttt{TSD}} \leftarrow \pi_\theta(o, t_\star \hat{a}_0^{\texttt{TSD}} + (1 - t_\star)z, t_\star)$.

**Minimal Iterative Policy.** We find that $\pi^{\texttt{TSD}}$ performs equivalently to a minimal policy which only adds training noise in the second step and has no stochasticity at inference time, which we call the minimal iterative policy.

> **Minimal Iterative Policy (MIP; ours)**
>
> Minimal Iterative Policy (**MIP**), representing Components 2 and 3, is trained via
>
> $$\pi_\theta^{\texttt{MIP}} \approx \arg\min_\theta \mathbb{E}(\|(\pi_\theta(o, I_0 = 0, t = 0) - a)\|^2 + \|(\pi_\theta(o, I_{t_\star}, t_\star) - a)\|^2), \quad (4.2)$$
>
> where $(o, a) \sim p_{\text{train}}, z \sim \mathrm{N}(0, \mathbf{I}), t_\star := .9$. At inference time, we compute:
>
> $$\hat{a}_0^{\texttt{MIP}} \leftarrow \pi_\theta^{\texttt{MIP}}(o, 0, t = 0), \quad \hat{a}^{\texttt{MIP}} \leftarrow \pi_\theta^{\texttt{MIP}}(o, t_\star \hat{a}_0^{\texttt{MIP}}, t_\star). \quad (4.3)$$

Minimal iterative policy provides a *minimal* implementation that still exhibits competitive performance with flow. Starting, with **TSD** and replace $(t_\star)^{-1} I_{t_\star}$ in the first term of the loss in Eq. (4.1) with its expectation $a = (t_\star)^{-1} \mathbb{E}[I_{t_\star}]$. We set the initial noise $I_0 = 0$ to be zero, so that $z$ only contributes to the second training loss. Finally, we sample with $z = 0$ to isolate the effect of adding stochasticity at training time, without stochasticity at inference time (c.f. Table 1). Since we provide supervision for both first step $\pi_\theta^{\texttt{MIP}}(o, I_0 = 0, t = 0)$ and second step $\pi_\theta^{\texttt{MIP}}(o, I_0 = I_{t_\star}, t = t_\star)$ with ground truth action $a$, **MIP** also exemplifies SIC in its simplest form. We compare **MIP** to Shorctu Models in Section D.8.

**Additional methods.** Straight Flow (**SF**, ours), representing only Component 2, further simplifies MIP to a single stage by setting the interpolation index $t^* = 1$ and removing the second term: $\pi_{\texttt{SF}_\theta} \approx \mathbb{E}\left[\|\pi_\theta(o, z, t = 0) - a\|^2\right]$. Finally, we study residual regression (**RR**), which replaces $I_{t_\star}$ in Eq. (4.2) with its expectation over $z$: $\mathbb{E}[I_{t_\star}] = t_\star a$. This preserves SIC (Component 3) yet removes stochasticity injection. Full details are provided in Section C.

To summarize, minimal iterative policy (**MIP**), straight-flow (**SF**) and residual regression (**RR**) represent all combinations of Components 2 and 3 without exhibiting Component 1.

## 4.2 COMPONENTS 2 AND 3 DRIVE PERFORMANCE: MIP MATCHES FLOW

Based on the design space parsing in Section 4, we are able to systematically ablate different design components' contribution to the final performance in Fig. 4 and Table 9. Our evaluation shows that either stochasticity injection (Component 2, exhibit by **SF**) or supervised iterative computation (Component 3, exhibited by **RR**) in isolation do not match the success of GCPs. **MIP**, being the

---

[1]Note that Component 1 refers to *training* a model to fit a conditional distribution, not necessarily to the sampling. For example, training $b_\theta$ via flow model but conducting deterministic inference with $\Phi_{\theta, \text{eul}}(z = 0 \mid o)$ is still considered distributional learning.

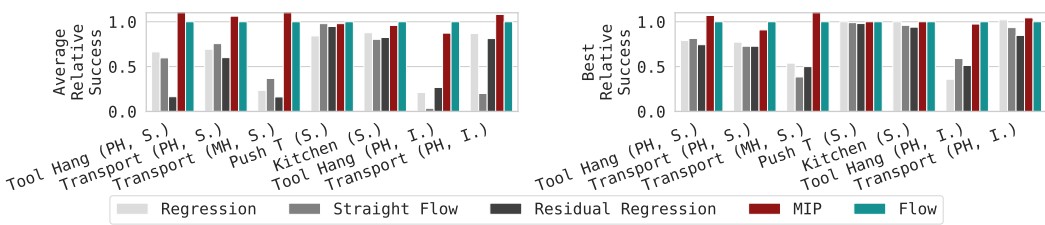

**Figure 4: Performance comparison between MIP and its variants on single-task benchmarks.** Average relative success rate on worst architecture and the best relative success rate on optimal architecture are reported. "*S*": state; "*I*": image.

only method which combines *supervised* iterative computation and stochasticity injection, achieves success on par with flow. Thus we conclude: the performance of GCPs comes from combining stochastic injection and iterative computation. Distributional training appears to be the least important factor. To further rule out the effect of distributional training and demonstrate the computation efficiency of MIP, we compare it with consistency models in Section D.8, where we find that MIP matches the performance of common consistency models with half of the training time.

**Remark 4.1.** Section C.3 exhibits two further variants which preserve Components 2 and 3: one that does not supervise intermediate steps, and a second which does not condition a time step $t_\star$. The latter does not enable network to learn separate functions across time steps. Both perform even worse than regression, confirming the importance of supervision of intermediate steps and decoupling network behavior across time steps.

## 5 INDUCTIVE BIAS, NOT EXPRESSIVITY, EXPLAINS MIP'S PERFORMANCE

### 5.1 MANIFOLD ADHERENCE, NOT RECONSTRUCTION, DRIVES PERFORMANCE

MIP, and the absence of multimodality, suggest a better ability to approximate the expert more accurately on training data. We test this by evaluating the $L_2$-error, i.e., reconstruction error, on validation set. Surprisingly, we find that MIP, Flow, and RCP exhibit the *same* validation loss; hence validation loss does predict their relative performance. Section G.1 reveals that validation loss doesn't correlate with performance across other axes of variation. Indeed, policy performance requires taking good actions on *o.o.d. states* under compounding error at deployment time (Simchowitz et al., 2025).

| Metric | Regression | SF | RR | MIP | Flow |
|---|---|---|---|---|---|
| Off-manifold $L_2$ | 0.067 | 0.063 | 0.062 | 0.054 | 0.042 |
| Validation $L_2$ | 0.290 | 0.234 | 0.224 | 0.195 | 0.217 |

**Table 4: Comparison of different methods on manifold adherence and reconstruction error.** Results are averaged across 3 different architectures and 32 states on state-based Tool-Hang with deterministic dataset.

Thus, we study a proxy which reflects performance in o.o.d. situations. We perturb expert trajectories in dataset as described in Section E.1, and evalute a novel metric that we call the *off-manifold norm*. Informally, this measures the projection error of a predicted action $a$ onto the space spanned by expert actions at neighboring states; see Section G.2 a for formal definition. Our metric assesses the quality of actions under simulated compounding error. Table 4 reports both $L_2$ validation loss and off-manifold $L_2$ norm for different methods: while all methods achieve low validation loss, only MIP and Flow are able to achieve low off-manifold $L_2$ norm, indicating their better manifold adherence. As SF does not exhibit the same benefit, we conclude that supervised iterative computation facilitates projection onto the manifold of expert actions by refining the prediction across sequential steps. Section L provides additional confirmation of this hypothesis on comprehensive toy experiments: GCPs are no better than RCP at fitting high frequency functions, but exhibit lower on-manifold error, suitably defined.

**Why manifold adherence matters for control.** We conjecture that, for high-precision tasks, the sensitivity to errors is not homogeneous across error directions in action space. Our findings present preliminary evidence that some form an "on-manifold inductive bias" directly aligns with minimizing error along relevant directions, yet is permissive to error in directions of lesser consequence. We think that rigorously establishing this hypothesis is an exciting direction for future work.

**No known mechanism accounts for greater manifold adherence in GCPs vs. RCPs.** There is a growing body of literature that shows that, if training data are supported on a given low dimensional manifold $\mathcal{M}$, then generative models learn to project onto $\mathcal{M}$ (Boffi et al., 2024; Permenter & Yuan, 2024). However, to our knowledge, there is no work that explains why this inductive bias would be *stronger* than what would be achieved with a well-trained regression model. Specifically, if $o \mid a$ lies in some (local) manifold, regression too should learn to project onto it.

One might conjecture that the iterative computation provides many changes to predict an action that "stick" to the action manifold. However, such a mechanism would require that once an on-manifold action is predicted, subsequent predictions do not nudge the prediction off-manifold. In Section K, we show that simple arguments based on implicit regularization in linear models do not suffice to explain this hypothesis, at least for **MIP**. Much like the usefulness of manifold adherence for control described above, the mechanism behind manifold adherence remains a mystery for future study.

## 5.2 STOCHASTICITY STABILIZES ITERATIVE COMPUTATION

We recall from Figure 4 that **SF** matches regression, whilst **RR** under-performs regression. This suggests that sequential action generation is highly brittle in the absence of stochasticity (Permenter & Yuan, 2024). Our findings support the hypothesis that stochasticity injection serves to provide "coverage" of the generative process. Note that this is different from task MDP-level augmentation like image augmentation or exploratory data collection since the augmentation happens in iterative generative process. Specifically, we can think of learning to perform two-stage action generation as an "internal" behavior cloning problem (Ren et al., 2024) under the dynamics induced by the generative process. Injecting stochasticity amounts to enhancing coverage of the action $\hat{a}_0$ in the first step of **MIP**, thus enable iterative improvement with more NFEs (Section D.7). Its benefits are analogous to trajectory noising effective in other behavior cloning applications (Laskey et al., 2017; Block et al., 2023; 2024; Simchowitz et al., 2025; Zhang et al., 2025). Similar benefits are found in the improved sensitivity analysis of diffusion relative to flows (Albergo et al., 2024).

## 5.3 ARCHITECTURE REMAINS ESSENTIAL FOR SCALING

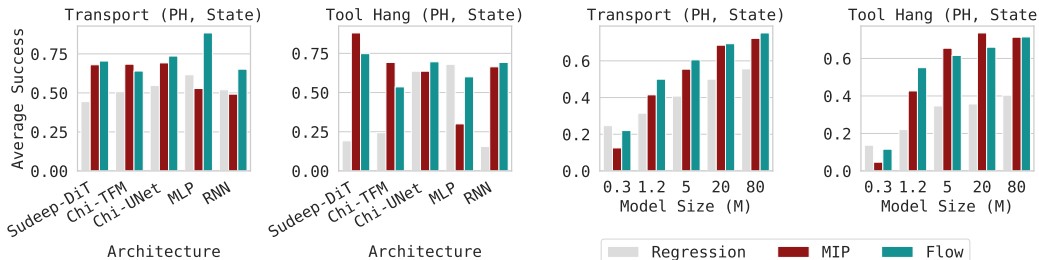

**Figure 5: Architecture and model size ablation.** Success rate are averaged across 3 seeds and 5 checkpoints on `Tool-Hang` and `Transport` tasks. Left 2 plots: architecture ablation. Right 2 plots: Model size ablation.

Regression, enjoys stronger relative performance at the smallest model sizes but scales more poorly than flow and **MIP** with increased model capacity (Fig. 5). We conjecture that supervised iterative computation can better utilize larger models, both by introducing more supervision steps at training, and by providing more parameters to represent different computations at successive generation steps. Nevertheless, *architecture design* plays an incredibly significant role. To showcase its importance, we ablate the performance of different method's average performance across both the 3 architectures above, and the more traditional MLP and RNN architectures, implemented with modern best practices including FiLM conditioning (Perez et al., 2018), and skip-connections (He et al., 2016)/LayerNorm (Ba et al., 2016) where appropriate (details in Section D.2). As demonstrated in Fig. 5, the combination of training method and architecture design has a strong yet somewhat erratic effect on both GCPs and RCP performance. In `Tool-Hang`, RCP achieves the best performance with an MLP architecture. In `Transport`, MLP with flow can even outperform more expressive architectures like `Chi-Transformer`. The coupling between training and architecture choice highlights the importance of controlling architecture design when comparing across methods.

ACKNOWLEDGEMENTS

Max Simchowitz and Giri Anantharaman acknowledge a TRI University 2.0 Fellow and Google Robotics Research Award. Max Simchowitz and Chaoyi Pan thank Nur Muhummad (Mahi) Shuffiulah for his insightful feedback, and Max Simchowitz also thanks Aviral Kumar, Sarvesh Patil, and Andrej Risteski for their thoughtful suggestions. Guannan Qu acknowledges support from NSF CAREER Award 2339112, NSF Award 2512805, and the Pennsylvania Infrastructure Technology Alliance. Guanya Shi holds concurrent appointments as an Assistant Professor at Carnegie Mellon University and as an Amazon Scholar. This paper describes work performed at Carnegie Mellon University and is not associated with Amazon.

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

# A  RELATED WORK

**Robotic Behavior Cloning.**  Behavior cloning (BC), also known as learning from demonstrations (LfD), has become a popular paradigm to enable robots to conduct complex, diverse and long-horizon manipulation tasks by learning from expert demonstrations (Argall et al., 2009; Zhu & Hu, 2018; Zhao et al., 2023b; Chi et al., 2024; Lin et al., 2024). In parallel, "robot foundation models" scale BC with internet-pretrained vision-language transformer-based backbones (Brohan et al., 2022; Zitkovich et al., 2023; O'Neill et al., 2024) and large-scale teleoperation datasets (Kim et al., 2024; Team et al., 2024). More recently, to better model continuous actions, generative models like diffusion and flow have been adopted to replace the tokenization method in transformers to achieve more expressive policies (NVIDIA et al., 2025; Black et al., 2024; Physical Intelligence et al., 2025; Liu et al., 2024). This work focuses on the generative modeling part of the behavior cloning pipeline, ablating the key design choices that lead to the success of generative control policies.

**Generative Modeling.**  The recent success of behavior cloning policies is built upon a rapid evolution of generative modeling techniques, starting from tokenization methods (Brown et al., 2020; Chen et al., 2021; Pertsch et al., 2025) and adversarial methods (Brock et al., 2019; Goodfellow et al., 2020; Ho & Ermon, 2016). Later, probabilistic generative models with iterative computation like diffusion models (Ho et al., 2020; Song et al., 2021b; Lu et al., 2025; Song et al., 2022; Nichol & Dhariwal, 2021; Karras et al., 2022) became a popular choice for generative modeling thanks to their better training stability and sampling quality. Flow models (Lipman et al., 2023; Albergo & Vanden-Eijnden, 2022; Liu et al., 2022) and consistency/shortcut models (Song et al., 2023; Song & Dhariwal, 2023; Meng et al., 2023; Boffi et al., 2025a; Geng et al., 2025) were later developed to achieve faster sampling while maintaining the expressivity of diffusion models. Though there have been extensive studies on probabilistic generative modeling's effectiveness in image and text generation (Lee et al., 2023; Chen et al., 2023), its mechanism in control, especially the key design choices, are still opaque in decision making.

**Generative Control Policies.**  To model diverse and complex behaviors, GCPs parameterize the relationship between observations and actions as a distribution rather than a deterministic function. Early works use transformers with tokenizers (Chen et al., 2021; Shafiullah et al., 2022), energy functions (Florence et al., 2022; Dasari et al., 2024) and VAEs (Zhao et al., 2023b) to parameterize the distribution. Diffusion models (Reuss et al., 2023; Chi et al., 2023; Ke et al., 2024; Dong et al., 2024; Janner et al., 2022; Yang et al., 2024) were introduced for their better expressivity of complex and multi-modal behaviors, followed by flow-based (Zhang et al., 2024; Black et al., 2024; Physical Intelligence et al., 2025) and flow-map/consistency-model/shortcut-model-based acceleration methods (Hu et al., 2024; Prasad et al., 2024; Sheng et al., 2025).

**Theoretical Literature on GCPs.**  Block et al. (2024) established that GCPs can imitate arbitrary expert distributions. Given our findings on the absence of multi-modality, a more closely related theoretical findings is that of Simchowitz et al. (2025), which elucidates how GCPs can circumvent certain worst-case compounding error phenomena in continuous-control imitation learning. Though the proposed mechanism is different, that finding is conceptually similar to our own: GCPs benefits arise from their favorable out-of-distribution properties, rather than raw expressivity of fitting in-distribution expert behavior.

## A.1  PREVIOUS WORKS' CONNECTION WITH GCP'S TAXONOMY.

We classify GCPs into three components: distributional learning, stochasticity injection, and supervised iterative computation. Starting from regression, it has none of the three components. To model a more complex distribution, Gaussian Mixture Model (GMM) (Zhu & Hu, 2018) was used to parameterize the distribution, trained with cross entropy loss. To make the network be able to represent more complex distirbutions, prior to diffusion, non-parametric method like VAEs (Zhao et al., 2023b) was used to parameterize the distribution, trained with reconstruction loss. During the training, a latent variables is predicted to predict the style the motion by mapping it from a noise $z$. Another line of work try to improve the policy expressivity by introducing iterative compute, like implicit behavior cloning (Florence et al., 2022; Dasari et al., 2024) and behavior transformer (Shafiullah et al., 2022). In IBC, the idea is to allow the network predict the energy function of the action rather the action itself. Compared to diffusion, the major difference is that they do not explicitly injecting noise during training and no intermediate supervision is provided for the intermediate re-

sults. Similarly, in behavior transformer, a two step policy is introduced to first predict the policy class and then refine it with another network to achieve higher precision control. Lastly, flow-based GCPs (Zhang et al., 2024; Black et al., 2024; Physical Intelligence et al., 2025), which holds all the three components and demonstrate state-of-the-art performance on popular benchmarks. In this paper, we look into a new combination that haven't been explored before, which is the combination of stochasticity injection and supervised iterative computation.

## B    LEARNING DYNAMICS OF GCPs AND RCPs GIVEN DIFFERENT TYPES OF DATA

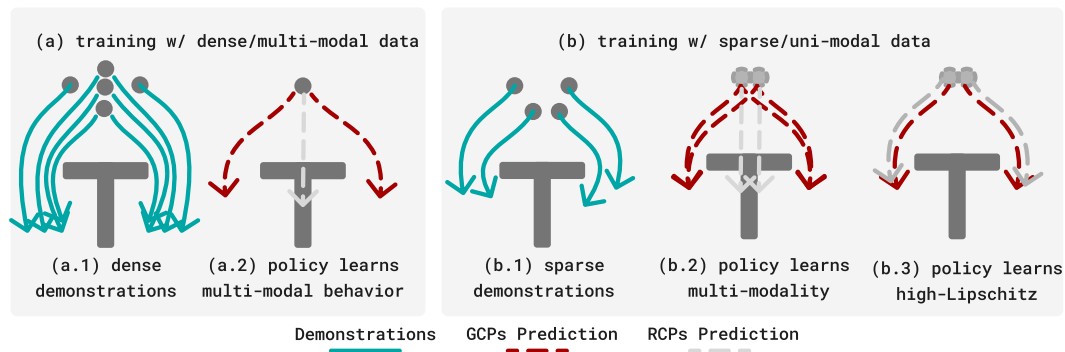

Figure 6: **GCP behavior given different types of data.** (a) Given true multi-modal data (a.1), GCPs learn both modes while RCPs collapse (a.2). (b) Given sparse data (b.1) in high-dimensional space, both policies can learn high-Lipschitz policies to quickly switch between modes (b.3). We find both **GCPs and RCPs** learn (b.3) in high-dimensional tasks (Section 3.2). Theorem 1 suggests that GCPs have a limited advantage over RCPs from a pure expressivity perspective.

## C    ADDITIONAL POLICY PARAMETRIZATIONS

This section further elaborates the design space of **MIP** in stochasticity injection, iterative computation and intermediate supervision.

### C.1    FULL ABALATION OF **MIP** VARIANTS

This section formally describes the training process of all **MIP** with different stochasticity injection and supervised iterative computation design.

**Residual Regression (RR)**    removes all stochasticity in training and the training objective is:

$$\pi_\theta^{\mathbf{RR}} \approx \arg\min_\theta \mathbb{E}_{(o,a)\sim p_{\text{train}}, z\sim N(0,\mathbf{I})} \tag{C.1}$$

$$\left( \|(\pi_\theta(o, I_0 = 0, t = 0) - t_\star a)\|^2 + \|(\pi_\theta(o, \text{sg}(\pi_\theta(o, I_0 = 0, t = 0)), t_\star) - a)\|^2 \right). \tag{C.2}$$

**Two-Step Denoising (TSD)**    The training objective is:

$$\pi_\theta^{\mathbf{TSD}} \approx \arg\min_\theta \mathbb{E}_{(o,a)\sim p_{\text{train}}, z\sim N(0,\mathbf{I})}$$

$$\left( \|(\pi_\theta(o, I_0, t = 0) - t_\star a)\|^2 + \|(\pi_\theta(o, \text{sg}(\pi_\theta(o, I_0, t = 0)) + (1 - t_\star)z, t_\star) - a)\|^2 \right).$$

where $I_0 = z$. Compared to **MIP**, **TSD** adds stochasticity to both first step training.

**MIP with Data Augmentation (`MIP-Dagger`)** To understand the importance of decoupling for enabling iterative computation, we propose an additional variant of **MIP** that lies between **MIP** and **RR**, where the two steps are partially coupled. Since the training method of second iteration is similar to data augmentation, we call this variant **MIP-Dagger**:

$$\pi_\theta^{\texttt{MIP-Dagger}} \approx \arg\min_\theta \mathbb{E}_{(o,a)\sim p_{\text{train}}, z\sim \text{N}(0,\mathbf{I})}$$
$$(\|(\pi_\theta(o, I_0 = 0, t = 0) - t_\star a)\|^2 + \|(\pi_\theta(o, t_\star \text{sg}(\pi_\theta(o, I_0 = 0, t = 0)) + (1 - t_\star)z, t_\star) - a)\|^2),$$

where the major difference compared to **MIP** is the second step takes in the interpolant between first step output and noise rather than the action and noise.

**MIP without intermediate supervision (`MIP-NoSupervision`)** To understand the effect of intermediate supervision on iterative computation, we propose one variant of **MIP** that removes the supervision of intermediate computation steps while preserving stochasticity injection at training time, named **MIP-NoSupervision**:

$$\pi_\theta^{\texttt{MIP-NoSupervision}} \approx \arg\min_\theta \mathbb{E}_{(o,a)\sim p_{\text{train}}, z\sim \text{N}(0,\mathbf{I})}$$
$$(\|(\pi_\theta(o, t_\star \text{sg}(\pi_\theta(o, I_0 = 0, t = 0)) + (1 - t_\star)z, t_\star) - a)\|^2),$$

where the first step's output is unsupervised.

**MIP without $t$ conditioning** By removing $t$ conditioning in **MIP**, it degenerates to **SF**. Here we present the multi-step integration process for straight flow when action distribution is Dirac delta. The integrator from $s$ to $t$ is:

$$a_t = \frac{t - s}{1 - s}\pi_\theta(o, s \cdot a_s) + \frac{1 - t}{1 - s}a_s$$

### C.2 Additional Noise Injection Methods

While **MIP** only injects noise to action, we also explore the possibility of injecting noise to observation. We propose two variants of **MIP**: **MIP-Obs** and **MIP-Dagger-Obs**. In **MIP-Obs**, we perturb the first step's observation with noise $z$, while the second step's training is the same as the original **MIP** with decoupled training. In **MIP-Dagger-Obs**, we perturb the first step's observation with noise $z$, and the second step's training is conditioned on the first step's output, making it similar to Dagger. Major differnce compared to the original MIP: perturb the first step's observation. In both variants, we fixed $t_* = 0.9$ and all observation perturbation happens at observation embedding space with normalized features.

$$\pi_\theta^{\text{MIP-Obs}} \approx \arg\min_\theta \mathbb{E}_{\substack{(o+(1-t_*)z,a)\sim p_{\text{train}} \\ z\sim\mathcal{N}(0,I)}} \left[\|(\pi_\theta(o + (1-t_*)z, I_0 = 0, t = 0) - t_*a)\|^2\right.$$

$$\left. + \|(\pi_\theta(o, I_{t_*}, t_*) - a)\|^2\right]$$

$$\pi_\theta^{\text{MIP-Dagger-Obs}} \approx \arg\min_\theta \mathbb{E}_{\substack{(o,a)\sim p_{\text{train}} \\ z\sim\mathcal{N}(0,I)}} \left[\|(\pi_\theta(o + (1-t_*)z, I_0 = 0, t = 0) - t_*a)\|^2\right.$$

$$\left. + \|(\pi_\theta(o, t_*\text{stopgrad}(\pi_\theta(o + (1-t_*)z, I_0 = 0, t = 0)) + 1 - t_*)z, t_*) - a)\|^2\right]$$

We find that perturbing observations introduces data conflicts and degrades performance (Table 5). In a two-step model, selecting noise levels that prevent observation overlap becomes challenging and brittle, leading to training instability across architectures.

| Architecture | Method (L2) | Transport (ph) | Tool-Hang (ph) |
|---|---|---|---|
| Chi-Transformer | **Regression** | 0.50/0.45 | 0.50/0.37 |
| Chi-Transformer | **MIP** | 0.79/0.69 | 0.92/0.85 |
| Chi-Transformer | **MIP-Dagger-Obs** | 0.00/0.00 | 0.00/0.00 |
| Chi-Transformer | **MIP-Obs** | 0.61/0.46 | 0.13/0.08 |
| Chi-Transformer | **Flow** | 0.81/0.71 | 0.89/0.75 |
| Sudeep-DiT | **Regression** | 0.65/0.54 | 0.31/0.25 |
| Sudeep-DiT | **MIP** | 0.80/0.69 | 0.80/0.72 |
| Sudeep-DiT | **MIP-Dagger-Obs** | 0.00/0.00 | 0.00/0.00 |
| Sudeep-DiT | **MIP-Obs** | 0.00/0.00 | 0.00/0.00 |
| Sudeep-DiT | **Flow** | 0.79/0.65 | 0.73/0.61 |
| Chi-UNet | **Regression** | 0.66/0.59 | 0.73/0.59 |
| Chi-UNet | **MIP** | 0.81/0.72 | 0.82/0.71 |
| Chi-UNet | **MIP-Dagger-Obs** | 0.00/0.00 | 0.00/0.00 |
| Chi-UNet | **MIP-Obs** | 0.00/0.00 | 0.00/0.00 |
| Chi-UNet | **Flow** | 0.83/0.75 | 0.87/0.73 |

**Table 5:** Performance comparison of different methods with observation perturbation on state-based tasks. For each methods and architecture, we report the average and best performance across 5 checkpoints with 3 random seeds.

## C.3 EXPERIMENT RESULTS

We benchmark all methods on the `Tool-Hang` task, given it is the one with the largest gap between RCP and GCPs. From Table 6, we can see that the important part is to add stochasticity injection between two iterations, and intermediate supervision is also important to realize the potential of iterative computation.

| Method | NFEs | Success Rate |
|---|---|---|
| **TSD** | 2 | 0.80 |
| **MIP** | 2 | 0.80 |
| **MIP-NoSupervision** | 2 | 0.42 |
| **MIP-Dagger** | 2 | 0.64 |
| **RR** | 2 | 0.54 |
| **SF** | 1 | 0.54 |
| **SF** | 3 | 0.55 |
| **SF** | 9 | 0.52 |

**Table 6:** Success rates across different **MIP** variants and **RR** on `Tool-Hang` task over 5 checkpoints across 3 architectures.

## D CONTROL EXPERIMENTS

### D.1 TASK SETTINGS

This section introduces all the tasks presented in the main paper. To reach a sound conclusion, use common benchmarks appears in previous works:

**Robomimic** Robomimic (Mandlekar et al., 2021) is a large-scale robotic manipulation benchmark designed to study imitation learning and offline reinforcement learning. It contains five manipulation tasks (`Lift`, `Can`, `Square`, `Transport`, `Tool-Hang`) with *proficient human (PH)* teleoperated demonstrations, and for four of them, additional *mixed proficient/non-proficient human*

*(MH)* demonstration datasets are provided (9 variants in total). We report results on both *state-based* and *image-based* observations, since these two modalities pose distinct challenges. Among the tasks, `Tool-Hang` requires extremely precise end-effector positioning and fine-grained contact control, while `Transport` demands high-dimensional control and coordination over extended horizons.

**Push-T** `Push-T` (Florence et al., 2022) is adapted from the Implicit Behavior Cloning (IBC). The task involves pushing a T-shaped block to a fixed target location using a circular end-effector. Randomized initializations of both the block and the end-effector introduce significant variability. The task is contact-rich and requires modeling complex object dynamics for precise block placement. Two observation variants are considered: (*i*) raw RGB image observations and (*ii*) state-based observations containing object pose and end-effector position.

**Kitchen** The Franka `Kitchen` environment is designed to test the ability of IL and offline RL methods to perform long-horizon, multi-task manipulation. It includes 7 interactive objects, with human demonstration data consisting of 566 sequences, each completing 4 sub-tasks in arbitrary order (e.g., opening a cabinet, turning a knob). Success is measured by completing as many of the demonstrated sub-tasks as possible, regardless of order. This setup explicitly introduces both short-horizon and long-horizon multimodality, requiring policies to generalize across compositional tasks.

**MetaWorld** `MetaWorld` is a large-scale suite of diverse manipulation tasks built in MuJoCo, where agents must perform challenging object interactions using a robotic gripper. We adopt the 3D observation setting using point cloud representations, ported from the DP3 framework (Ze et al., 2024), to better evaluate geometric reasoning and spatial generalization. Tasks in MetaWorld are categorized into different difficulty levels, with benchmarks testing few-shot adaptation and multi-task transfer learning.

**Adroit** `Adroit` is a suite of dexterous manipulation tasks featuring a 24-DoF anthropomorphic robotic hand. Tasks include pen rotation, door opening, and object relocation, all of which demand precise, coordinated multi-finger control. Following DP3 (Ze et al., 2024), we use point cloud observations to capture fine-grained 3D object-hand interactions. Policies are trained using VRL3, highlighting the challenges of high-dimensional control and sim-to-real transfer in dexterous manipulation.

**LIBERO** `LIBERO` is a common multi-task benchmark to evaluate VLA's generalization ability. It is composed of 130 tasks and can be categorized into multiple categories, including object, goal, spatial, and 10-task. The 10-task is long horizon and considered the most challenging to solve.

## D.2 ARCHITECTURE DESIGN

We study four policy backbones—`Chi-Transformer`, `Sudeep-DiT`, `Chi-UNet`, `RNN`, and `MLP`—under a common training recipe and data interface. Unless otherwise specified, *all models are capacity-matched to ∼20M parameters* to enable fair comparison.

**Chi-UNet** is adopted from Diffusion Policy (Chi et al., 2023) which built on top of 1D temporal U-Net (Janner et al., 2022) with FiLM conditioning (Perez et al., 2018) on observation $o$ and flow time $t$. `Chi-UNet` has a strong inductive bias for the temporal structure of the action and tends to smooth out the action.

**Chi-Transformer** follows the time–series diffusion transformer from Diffusion Policy (Chi et al., 2023), where the noisy action tokens $a_t$ form the input sequence and a *positional embedding* of the flow time $t$ is prepended as the first token; observations $o$ are mapped by a shared MLP into an observation-embedding sequence that conditions the decoder stack. Compared to `Chi-UNet`, `Chi-Transformer` uses token-wise self-attention over the whole action sequence, thus can model less-smooth and more complex actions.

**Sudeep-DiT** is a DiT-style (Diffusion Transformer) conditional noise network specialized for policies adopted from DiT-Policy (Dasari et al., 2024): observation $o$ are first encoded into observation vectors; the flow time $t$ is embedded via *positional embedding*; an encoder–decoder transformer then fuses these with initial noise $z$ to predict next action. The key ingredient of `Sudeep-DiT` is replacing standard cross-attention with *adaLN-Zero* blocks—adaptive LayerNorm modulation using the mean encoder embedding and the time embedding, with zero-initialized output-scale projections—stabilizing diffusion training at scale. Compared to `Chi-Transformer`, `Sudeep-DiT` has adaLN-based conditioning (instead of vanilla cross-attention) and an explicit encoder-decoder split, yielding better training stability.

**RNN** The `RNN` backbone processes sequences with a stacked LSTM/GRU. For each action time step in the chunk, the input vector concatenates: the current noised action $a_t$, a time embedding for $t$, and a observation embedding for $o$. The RNN outputs are fed to a MLP head with LayerNorm+ApproxGELU+Dropout blocks before output the action with final linear head. All linear and recurrent weights use *orthogonal initialization* (biases zero), and RNN layer dropout is applied when depth>1.

**MLP** The `MLP` backbone flattens the action and observation, appending the time embedding. Each mlp block has LayerNorm, ApproxGELU and Dropout blocks with residual connection and *orthogonal* weight initialization throughout. Each block output is then modulated with FiLM conditioning.

**DP3** built on top of `Chi-UNet` with extra 3d perception encoder. We use the exact same architecture as 3D diffusion policy (Ze et al., 2024).

**Model hyperparameters** In the main experiments, we align the model capacity to 20M parameters for default if not specified, with detailed hyperparameters report in Table 7.

| Backbone | Heads | Layers | Embedding dim | Dropout |
|---|---|---|---|---|
| Sudeep-DiT | 8 | 8 | 256 | 0.1 |
| Chi-UNet | – | – | 256 | – |
| Chi-Transformer | 4 | 8 | – | 0.1 |
| RNN | – | 8 | 512 | 0.1 |
| MLP | – | 8 | 512 | – |

**Table 7:** Model hyperparameters.

### D.3 FINETUNING $\pi_0$ ON LIBERO

For $\pi_0$ finetuning experiments, we use `lerobot` framework (Cadene et al., 2024) to finetune $\pi_0$ on LIBERO. Our flow-based finetuning experiments match their reported results. To finetune $\pi_0$ to regression policy, we use the same architecture but set the initial noise always to zero and let the model directly predict the action. To finetune $\pi_0$ to **MIP**, we use the same practice where we modify the time step $t$ to be uniformly sample from $\{0, t^*\}$ uniformly and set the initial noise to zero. We train all policies until convergence with 50k gradient steps on 1 node with 8 H100 GPUs.

### D.4 FULL RESULTS FOR FLOW AND REGRESSION COMPARISON

In the paper, we only present the aggregated results across 3 architectures. Figure 7 present the full results across all architectures with different training methods.

To further rule out the effect of training method, we also compare different methods' performance with $\ell_1$, which is observed to be superior for regression policy (Kim et al., 2024). We also benchmark the performance of flow model and **MIP** with $\ell_1$ loss to understand the effect of loss function on the performance of GCPs. Table 8 shows the performance comparison of different methods with $\ell_1$ and $\ell_2$ loss, where we find that $\ell_1$ loss generally outperforms $\ell_2$ loss, especially for regression policy. We attribute the superior performance of $\ell_1$ loss to the fact that it can capture the expert behavior better by learning the medium instead of the mean of the action. However, even with $\ell_1$ loss, we still

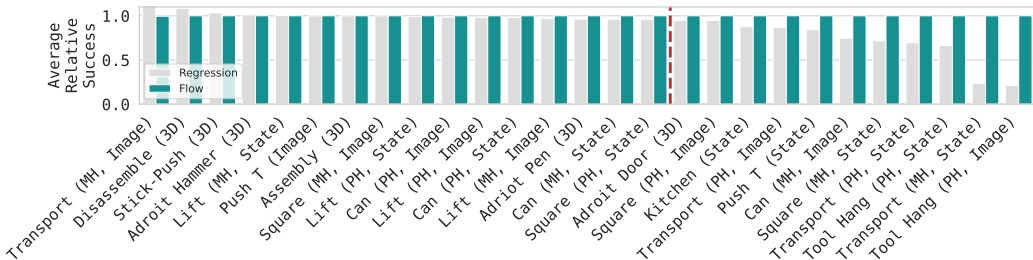

**Figure 7: Relative performance of RCP compared to GCP across common benchmarks (worst-case architecture).** For each task, we implement `Chi-Transformer`, `Sudeep-DiT` and `Chi-UNet`. For each architecture, we average performance of the last 5 training checkpoints across three seeds. We then report the performance of the worst-performing architecture, chosen individually for both RCP and GCP, to demonstrate method robustness. For **Flow**, we always do 9 step Euler integrations, where its performance plateaued. For readability, RCP success rates are plotted relative to flow, with flow normalized to performance of 1 per task. Tasks are grouped by observation modality, and ordered by relative RCP performance. Red dashed line indicates threshold at which RCP attains $< 95\%$ success of GCP.

observe that Regression $<$ **MIP** $\approx$ Flow, highlighting the importance of the stochasticity injection and iterative computation is independent of the loss function.

| Architecture | Method | Transport (ph) | Tool Hang (ph) |
|---|---|---|---|
| `Sudeep-DiT` | **Regression** $\ell_1$ | 0.72/0.64 | 0.76/0.65 |
| `Sudeep-DiT` | **Regression** $\ell_2$ | 0.50/0.45 | 0.50/0.37 |
| `Sudeep-DiT` | **MIP** $\ell_1$ | 0.81/0.73 | 0.91/0.84 |
| `Sudeep-DiT` | **MIP** $\ell_2$ | 0.79/0.69 | 0.92/0.85 |
| `Sudeep-DiT` | **Flow** $\ell_1$ | 0.83/0.76 | 0.93/0.84 |
| `Sudeep-DiT` | **Flow** $\ell_2$ | 0.81/0.71 | 0.89/0.75 |
| `Chi-Transformer` | **Regression** $\ell_1$ | 0.67/0.57 | 0.44/0.33 |
| `Chi-Transformer` | **Regression** $\ell_2$ | 0.65/0.54 | 0.31/0.25 |
| `Chi-Transformer` | **MIP** $\ell_1$ | 0.80/0.68 | 0.85/0.77 |
| `Chi-Transformer` | **MIP** $\ell_2$ | 0.80/0.69 | 0.80/0.72 |
| `Chi-Transformer` | **Flow** $\ell_1$ | 0.77/0.69 | 0.81/0.71 |
| `Chi-Transformer` | **Flow** $\ell_2$ | 0.79/0.65 | 0.73/0.61 |
| `Chi-UNet` | **Regression** $\ell_1$ | 0.84/0.71 | 0.71/0.55 |
| `Chi-UNet` | **Regression** $\ell_2$ | 0.66/0.59 | 0.73/0.59 |
| `Chi-UNet` | **MIP** $\ell_1$ | 0.85/0.68 | 0.76/0.67 |
| `Chi-UNet` | **MIP** $\ell_2$ | 0.81/0.72 | 0.82/0.71 |
| `Chi-UNet` | **Flow** $\ell_1$ | 0.85/0.69 | 0.87/0.71 |
| `Chi-UNet` | **Flow** $\ell_2$ | 0.83/0.75 | 0.87/0.73 |

**Table 8:** Comparison of $\ell_1$ vs $\ell_2$ norm across different methods and architectures. Report average/best performance across 5 checkpoints with 3 random seeds.

| **Method** | LIBERO Object | LIBERO Goal | LIBERO Spatial | LIBERO 10 |
|---|---|---|---|---|
| RCP ($\ell_2$ loss) | 92.6 | 94.6 | 97.2 | 78.0 |
| RCP ($\ell_1$ loss) | 95.2 | 88.0 | 95.8 | 62.4 |
| **Flow** | **97.4** | 95.0 | 95.8 | 81.6 |
| **MIP** | 95.8 | **95.2** | **97.6** | **82.2** |

**Table 9: Performance comparison on multi-task `LIBERO` benchmark.** We report the success rate of the checkpoint trained with 50k gradient steps of finetuning $\pi_0$ on the full `LIBERO` dataset. We implement **MIP** with $t^* = 0.9$ and integrate flow with 10 steps. For regression, we train with both $\ell_2$ and $\ell_1$ loss as suggested in (Kim et al., 2024).

## D.5 DATASET QUALITY ABLATION

GCPs are believed to handle data with diverse quality better. To test that assumption, we manually corrupt the expert dataset and inject stochactity and multi-modality in to the dataset. In Table 10, we compare 4 different datasets (3 of them collected by ourselves). In the collected dataset, we manually inject noise to the policy and add delay the policy from time to time to introduce multi-modality that is common in the real world.

| Architecture | Method | NFEs | Delayed & Noisy Policy (Worst Quality) | Delayed Policy (Mixed Quality) | Zero-Flow (Better Quality) | Proficient Human (Good Quality) |
|---|---|---|---|---|---|---|
| Chi-UNet | **Regression** | 1 | 0.70/0.63 | 0.80/0.72 | 0.76/0.65 | 0.76/0.62 |
| Chi-UNet | **SF** | 1 | 0.70/0.62 | 0.82/0.76 | 0.84/0.77 | 0.62/0.38 |
| Chi-UNet | **MIP** | 2 | 0.80/0.72 | 0.82/0.61 | 0.74/0.64 | 0.80/0.68 |
| Chi-UNet | **Flow** | 9 | 0.76/0.68 | 0.74/0.50 | 0.76/0.54 | 0.84/0.70 |
| Chi-Transformer | **Regression** | 1 | 0.38/0.22 | 0.40/0.31 | 0.42/0.26 | 0.50/0.24 |
| Chi-Transformer | **SF** | 1 | 0.46/0.35 | 0.68/0.50 | 0.56/0.41 | 0.62/0.48 |
| Chi-Transformer | **MIP** | 2 | 0.56/0.49 | 0.70/0.54 | 0.64/0.56 | 0.72/0.68 |
| Chi-Transformer | **Flow** | 9 | 0.56/0.34 | 0.54/0.48 | 0.62/0.49 | 0.68/0.54 |
| Sudeep-DiT | **Regression** | 1 | 0.42/0.29 | 0.36/0.28 | 0.42/0.32 | 0.30/0.19 |
| Sudeep-DiT | **SF** | 1 | 0.66/0.41 | 0.60/0.54 | 0.72/0.57 | 0.68/0.50 |
| Sudeep-DiT | **MIP** | 2 | 0.66/0.56 | 0.74/0.58 | 0.70/0.61 | 0.86/0.78 |
| Sudeep-DiT | **Flow** | 9 | 0.56/0.45 | 0.66/0.58 | 0.72/0.65 | 0.78/0.68 |

**Table 10:** Performance comparison across different methods and data quality levels. We evaluate on the task Tool-Hang with state observations using 10M parameter networks. Success rates are reported as averages over 5 checkpoints across 3 seeds.

## D.6 FULL RESULTS FOR **MIP** AND ITS VARIANTS

| Architecture | Method | Lift | | Can | | Square | | Transport | | Tool-Hang | Push-T | Kitchen |
|---|---|---|---|---|---|---|---|---|---|---|---|---|
| | | mh | ph | mh | ph | mh | ph | mh | ph | | | |
| Sudeep-DiT | Flow | **1.00**/0.99 | **1.00**/**1.00** | **1.00**/0.94 | **1.00**/1.00 | 0.88/0.75 | **1.00**/0.94 | 0.40/0.27 | 0.80/0.70 | 0.86/0.75 | **0.98**/**0.95** | 0.98/0.96 |
| Sudeep-DiT | Regression | **1.00**/0.99 | **1.00**/**1.00** | 0.92/0.90 | **1.00**/0.98 | 0.72/0.53 | 0.94/0.86 | 0.12/0.06 | 0.50/0.44 | 0.52/0.39 | 0.92/0.83 | 0.98/0.92 |
| Sudeep-DiT | Straight Flow | **1.00**/0.98 | **1.00**/**1.00** | 0.96/0.90 | **1.00**/0.99 | 0.72/0.66 | 0.96/0.93 | 0.20/0.14 | 0.56/0.48 | 0.70/0.59 | 0.90/0.86 | 0.96/0.91 |
| Sudeep-DiT | MIP | **1.00**/0.99 | **1.00**/**1.00** | 0.98/0.95 | **1.00**/1.00 | 0.90/0.81 | 0.98/**0.94** | 0.44/0.38 | 0.76/0.68 | **0.92**/**0.88** | 0.95/0.92 | **1.00**/**0.97** |
| Chi-Transformer | Flow | **1.00**/**1.00** | **1.00**/**1.00** | **1.00**/0.93 | **1.00**/0.98 | 0.78/0.74 | 0.96/0.89 | 0.44/0.34 | **0.88**/0.64 | 0.68/0.54 | 0.91/0.89 | **1.00**/0.96 |
| Chi-Transformer | Regression | **1.00**/0.99 | **1.00**/0.99 | 0.98/0.92 | **1.00**/0.96 | 0.74/0.61 | 0.92/0.85 | 0.28/0.20 | 0.68/0.51 | 0.40/0.36 | 0.93/0.88 | 0.98/0.91 |
| Chi-Transformer | Straight Flow | **1.00**/0.99 | **1.00**/**1.00** | 0.98/0.92 | **1.00**/0.99 | 0.68/0.58 | 0.96/0.89 | 0.24/0.16 | 0.62/0.54 | 0.60/0.55 | 0.94/0.90 | 0.96/0.92 |
| Chi-Transformer | MIP | **1.00**/**1.00** | **1.00**/**1.00** | 0.96/0.95 | **1.00**/1.00 | 0.86/0.73 | 0.96/0.89 | 0.42/0.37 | 0.80/0.68 | 0.76/0.69 | 0.94/0.92 | 0.98/0.96 |
| Chi-UNet | Flow | **1.00**/**1.00** | **1.00**/**1.00** | **1.00**/0.98 | **1.00**/0.99 | 0.90/0.78 | 0.98/0.94 | 0.52/0.40 | 0.80/**0.73** | 0.84/0.70 | **0.98**/0.94 | **1.00**/**0.97** |
| Chi-UNet | Regression | **1.00**/**1.00** | **1.00**/**1.00** | **1.00**/0.96 | **1.00**/0.99 | **0.94**/**0.82** | **1.00**/0.91 | 0.22/0.16 | 0.64/0.55 | 0.68/0.64 | 0.95/0.89 | 0.92/0.88 |
| Chi-UNet | Straight Flow | **1.00**/1.00 | **1.00**/**1.00** | **1.00**/0.92 | **1.00**/0.99 | **0.94**/0.79 | 0.98/0.90 | 0.22/0.15 | 0.64/0.52 | 0.50/0.00 | 0.93/0.88 | 0.86/0.79 |
| Chi-UNet | MIP | **1.00**/**1.00** | **1.00**/**1.00** | **1.00**/0.98 | **1.00**/0.99 | 0.92/0.81 | **1.00**/0.94 | **0.62**/**0.46** | 0.80/0.69 | 0.80/0.64 | 0.97/**0.95** | **1.00**/0.96 |

**Table 11:** Performance comparison of Flow and Regression methods across different **state-based** robotic manipulation tasks. For each task, we report the best checkpoint performance / averaged performance over last 5 checkpoints. Each experiment is run with 3 seeds and we report the average performance across all seeds.

| Architecture | Method | Lift | | Can | | Square | | Transport | | Tool Hang | PushT |
|---|---|---|---|---|---|---|---|---|---|---|---|
| | | mh | ph | mh | ph | mh | ph | mh | ph | | |
| Sudeep-DiT | Flow | **1.00**/**1.00** | **1.00**/1.00 | 0.96/0.94 | **1.00**/0.99 | 0.82/0.76 | 0.96/0.94 | 0.32/0.20 | 0.84/0.83 | **0.78**/0.57 | **0.92**/**0.89** |
| Sudeep-DiT | Regression | **1.00**/0.99 | **1.00**/**1.00** | 0.92/0.81 | **1.00**/**1.00** | 0.74/0.67 | 0.94/0.84 | 0.14/0.08 | 0.74/0.56 | 0.28/0.18 | 0.83/0.77 |
| Sudeep-DiT | Straight Flow | **1.00**/0.99 | **1.00**/0.99 | 0.98/0.95 | **1.00**/0.98 | 0.82/0.72 | **1.00**/0.93 | 0.26/0.19 | 0.86/0.83 | 0.46/0.40 | 0.85/0.79 |
| Sudeep-DiT | MIP | **1.00**/0.99 | **1.00**/**1.00** | **1.00**/0.96 | **1.00**/0.98 | 0.90/0.83 | **1.00**/0.92 | 0.50/0.31 | 0.90/0.84 | 0.76/**0.66** | 0.91/0.87 |
| Chi-Transformer | Flow | **1.00**/0.99 | **1.00**/1.00 | 0.98/0.92 | **1.00**/0.96 | 0.70/0.66 | 0.98/0.93 | 0.24/0.22 | 0.80/0.77 | 0.54/0.40 | 0.89/0.85 |
| Chi-Transformer | Regression | **1.00**/0.98 | **1.00**/1.00 | **1.00**/0.94 | **1.00**/0.96 | 0.76/0.70 | 0.98/0.90 | 0.40/0.27 | 0.94/0.87 | 0.44/0.36 | 0.85/0.81 |
| Chi-Transformer | Straight Flow | **1.00**/**1.00** | **1.00**/**1.00** | **1.00**/0.95 | **1.00**/0.98 | 0.90/0.78 | 0.98/**0.94** | 0.32/0.25 | 0.86/0.70 | 0.36/0.28 | 0.86/0.80 |
| Chi-Transformer | MIP | **1.00**/0.98 | **1.00**/**1.00** | 0.96/0.91 | **1.00**/0.96 | 0.72/0.21 | 0.90/0.04 | 0.18/0.06 | 0.86/0.69 | 0.60/0.48 | 0.87/0.83 |
| Chi-UNet | Flow | **1.00**/**1.00** | **1.00**/**1.00** | **1.00**/**0.97** | **1.00**/0.98 | 0.90/0.79 | 0.96/0.90 | 0.24/0.16 | 0.78/0.61 | 0.48/0.37 | **0.92**/0.87 |
| Chi-UNet | Regression | **1.00**/0.96 | **1.00**/0.99 | 0.84/0.70 | 0.98/0.87 | 0.74/0.64 | 0.94/0.86 | 0.18/0.10 | 0.66/0.64 | 0.30/0.23 | 0.90/0.85 |
| Chi-UNet | Straight Flow | **1.00**/0.94 | **1.00**/0.99 | 0.98/0.93 | **1.00**/0.96 | 0.72/0.68 | 0.92/0.62 | 0.00/0.00 | 0.50/0.22 | 0.06/0.02 | 0.87/0.82 |
| Chi-UNet | MIP | **1.00**/**1.00** | **1.00**/**1.00** | **1.00**/0.95 | **1.00**/0.98 | **0.92**/**0.84** | 0.96/0.91 | **0.52**/**0.37** | **0.96**/**0.91** | 0.56/0.50 | 0.83/0.78 |

**Table 12:** Performance comparison of Flow and Regression methods across different **image-based** robotic manipulation tasks. For each task, we report the best checkpoint performance / averaged performance over last 5 checkpoints. Each experiment is run with 3 seeds and we report the average performance across all seeds.

For Kitchen, the task has multiple stages. In the main results, we only report the performance of the last stage since it is the most challenging one. Table 14 shows the performance comparison across different design choices on Kitchen task.

| Architecture | Method | Adroit | | | MetaWorld | | |
|---|---|---|---|---|---|---|---|
| | | Hammer | Door | Pen | Stick-Push | Assembly | Disassemble |
| DP3 | Flow | $0.96 \pm 0.02$ | $\mathbf{0.60 \pm 0.06}$ | $\mathbf{0.54 \pm 0.11}$ | $0.92 \pm 0.04$ | $\mathbf{0.98 \pm 0.03}$ | $0.72 \pm 0.14$ |
| | Regression | $\mathbf{0.97 \pm 0.04}$ | $0.52 \pm 0.16$ | $0.47 \pm 0.08$ | $\mathbf{0.95 \pm 0.06}$ | $\mathbf{0.98 \pm 0.03}$ | $\mathbf{0.78 \pm 0.08}$ |

**Table 13:** Performance comparison of Flow and Regression methods using DP3 architecture across different **point-cloud-based** robotic manipulation tasks. For each task, we report the best checkpoint performance / averaged performance over last 5 checkpoints. Each experiment is run with 3 seeds and we report the average performance across all seeds.

| Architecture | Method | P1 | P2 | P3 | P4 |
|---|---|---|---|---|---|
| | **Flow** | 1.0 | 1.0 | 1.0 | 0.98 |
| Chi-UNet | **MIP** | 1.0 | 1.0 | 1.0 | 0.94 |
| | **Regression** | 0.98 | 0.94 | 0.94 | 0.86 |
| | **Flow** | 1.0 | 1.0 | 1.0 | 1.0 |
| Chi-Transformer | **MIP** | 1.00 | 0.98 | 0.98 | 0.96 |
| | **Regression** | 1.0 | 1.0 | 0.98 | 0.94 |
| | Flow | 1.0 | 1.0 | 1.0 | 0.98 |
| Sudeep-DiT | **MIP** | 1.00 | 1.00 | 1.00 | 0.98 |
| | **Regression** | 1.0 | 0.98 | 0.96 | 0.88 |

**Table 14: Performance comparison across different design choices on kitchen task.** Kitchen task has multiple stages and we report the success rate of finishing $n$ tasks in the table. For the performance reported in the main paper and previous tables, we report the success rate of finishing 4 tasks.

## D.7 DIFFERENT METHOD'S PERFORMANCE WITH DIFFERENT NUMBER OF FUNCTION EVALUATIONS

We also provide detailed evaluation on different method's scaling behavior given different amount of online computation budgets. Table 15 highlights that only **MIP** and **Flow** benefit from iterative computate.

| Method | Reg. | SF | | | RR | | MIP | | Flow | | |
|---|---|---|---|---|---|---|---|---|---|---|---|
| **NFEs** | 1 | 1 | 3 | 9 | 1 | 2 | 1 | 2 | 1 | 3 | 9 |
| **S.R.** | 0.46 | 0.54 | 0.55 | 0.52 | 0.31 | 0.33 | 0.50 | 0.74 | 0.32 | 0.55 | 0.66 |

**Table 15: Comparison of methods and their corresponding number of function evaluations (NFEs).** Evaluated on state-based `Tool-Hang` task over `Chi-UNet`. Average success rate is reported across 3 seeds and 5 checkpoints.

## D.8 COMPARING MIP WITH CONSISTENCY MODELS

Given **MIP** takes less integration steps compared to flow model, we compare it with consistency models which accelerate the sampling process of flow by distilling the learned flow into a shortcut model. The major difference between **MIP** and consistency models is that the latter do satisfy C1, and require training over a continuum of noise levels. On the other hand, **MIP** is trained to predict the conditional mean of the interpolant, and thus, doesn't need extra distillation stage. As shown in Table 16, We benchmarks **MIP** to common consistency model training methods including consistency trajectory model (CTM) (Kim et al., 2023) and Lagrangian map distillation (LMD) (Boffi et al., 2025a), where LMD only works for `Chi-UNet` due to its dependency on jacobian matrix computation. The benchmarking results indicates that, given best architecture, **MIP** outperforms consistency models. In terms of training time, **MIP** only takes half of the time compared to CTM, where LMD training takes even longer due to jacobian matrix computation.

| Architecture | Method | Transport | | Tool Hang |
|---|---|---|---|---|
| | | mh | ph | |
| `Sudeep-DiT` | Flow | 0.40/0.27 | 0.80/0.70 | 0.86/0.75 |
| `Sudeep-DiT` | MIP | 0.44/0.38 | 0.76/0.68 | **0.92/0.88** |
| `Chi-Transformer` | CTM | 0.57/0.32 | **0.90**/0.58 | 0.56/0.26 |
| `Chi-Transformer` | Flow | 0.44/0.34 | 0.88/0.64 | 0.68/0.54 |
| `Chi-Transformer` | MIP | 0.42/0.37 | 0.80/0.68 | 0.76/0.69 |
| `Chi-UNet` | CTM | 0.40/0.32 | 0.72/0.63 | 0.46/0.37 |
| `Chi-UNet` | Flow | 0.52/0.40 | 0.80/**0.73** | 0.84/0.70 |
| `Chi-UNet` | LMD | 0.44/0.32 | 0.76/0.68 | 0.74/0.52 |
| `Chi-UNet` | MIP | **0.62/0.46** | 0.80/0.69 | 0.80/0.64 |

**Table 16:** Benchmark results across different architectures and methods on state-based tasks on consistency models and `MIP`. Report average/best performance across 5 checkpoints with 3 random seeds. Both LMD and CTM integrate 2 steps, which is the same as `MIP`.

# E  LIPSCHITZ CONSTANT STUDY DETAILS

## E.1  LIPSCHITZ EVLUATION METHOD

We note that not all inputs $o$ are dynamically feasible, and our dataset lies only on a narrow manifold of the observation space. Therefore, we must carefully evaluate the Lipschitz constant on the feasible observation space to avoid conflating model expressivity with errors arising from infeasible states. To ensure feasibility, instead of directly perturbing the state, we perturb the action and then roll it out in the environment. This guarantees that both the perturbed state and the resulting observation remain feasible.

In practice, we identify states that exhibit the highest ambiguity of actions in the dataset, referred to as *critical states*. For each critical state, we inject Gaussian noise $\eta \sim \mathcal{N}(0, \epsilon^2 I)$ into the normalized action, unnormalize it, and then roll it out. We select $100$ critical states from the dataset. For each state, we perturb the corresponding expert action $a$ with $64$ independent Gaussian samples.

Let $o$ denote the next nominal observation after applying the nominal action $a$. After rolling out the perturbed actions, we obtain perturbed observations $o^{(1)}, \ldots, o^{(N_{\text{perturb}})}$. The policy then predicts the perturbed actions $a^{(i)} = \pi(o^{(i)})$. To ensure comparability across different states and tasks, we evaluate the Lipschitz constant with respect to normalized observations $\bar{o} = \frac{o - \mu_o}{\sigma_o}$ and normalized actions $\bar{a} = \frac{a - \mu_a}{\sigma_a}$. Finally, the Lipschitz constant is estimated using a zeroth-order approximation:

$$L \approx \max_i \frac{\|\bar{a}^{(i)} - \bar{a}\|_2}{\|\eta\|_2}. \tag{E.1}$$

Full version of above process is stated in Algorithm 1.

---

**Algorithm 1** Lipschitz Constant Estimation via Action Perturbation

---

**Require:** Dataset $\mathcal{D}$, policy $\pi$, noise scale $\epsilon$, number of critical states $N_s{=}100$, number of perturbations $N_p{=}64$
**Ensure:** Estimated Lipschitz constant $L$
1: $S \leftarrow$ identify $N_s$ critical states from $\mathcal{D}$        ▷ Select states with highest action ambiguity
2: **for all** critical state $s \in S$ **do**
3:     $(a, o) \leftarrow$ expert action and nominal next observation for $s$       ▷ Get ground truth action-observation pair
4:     $(\bar{a}, \bar{o}) \leftarrow$ normalize $(a, o)$ using dataset statistics ▷ Ensure comparability across states/tasks
5:     **for** $i = 1$ to $N_p$ **do**
6:        $\eta \sim \mathcal{N}(0, \epsilon^2 I)$                       ▷ Sample Gaussian perturbation
7:        $a_{\text{pert}} \leftarrow$ unnormalize$(\bar{a} + \eta)$       ▷ Create perturbed action in original scale
8:        $o^{(i)} \leftarrow$ rollout$(a_{\text{pert}})$ in environment    ▷ Execute perturbed action to get feasible state
9:        $\bar{o}^{(i)} \leftarrow$ normalize$(o^{(i)})$              ▷ Normalize perturbed observation
10:       $a^{(i)} \leftarrow \pi(o^{(i)})$              ▷ Get policy prediction on perturbed state
11:       $\bar{a}^{(i)} \leftarrow$ normalize$(a^{(i)})$             ▷ Normalize predicted action
12:       $r_i \leftarrow \frac{\|\bar{a}^{(i)} - \bar{a}\|_2}{\|\eta\|_2}$         ▷ Compute finite difference approximation
13:     $L_s \leftarrow \max_i r_i$              ▷ Local Lipschitz constant for state $s$
14: $L \leftarrow \frac{1}{N_s} \sum_{s=1}^{N_s} L_s$           ▷ Average across all critical states
15: **return** $L$

---

### E.2 FULL LIPSCHITZ EVALUATION RESULTS

In the main text, we only report the average Lipschitz constant on critical states across 3 architectures. Here, we report the full Lipschitz constant evaluation reuslt in Table 17 with different architectures and tasks.

| Task | Architecture | Method | Lipschitz Constant (Policy) |
|---|---|---|---|
| Push-T (State) | Chi-UNet | **Regression** | $0.85 \pm 0.58$ |
| | | **Flow** | $0.31 \pm 0.01$ |
| | Sudeep-DiT | **Regression** | $0.52 \pm 0.11$ |
| | | **Flow** | $0.22 \pm 0.02$ |
| | Chi-Transformer | **Regression** | $1.33 \pm 1.14$ |
| | | **Flow** | $0.82 \pm 0.26$ |
| Kitchen (State) | Chi-UNet | **Regression** | $13.47 \pm 2.80$ |
| | | **Flow** | $13.31 \pm 4.13$ |
| | Sudeep-DiT | **Regression** | $15.37 \pm 3.69$ |
| | | **Flow** | $12.54 \pm 5.09$ |
| | Chi-Transformer | **Regression** | $13.37 \pm 4.00$ |
| | | **Flow** | $11.44 \pm 4.10$ |
| Tool-Hang (PH, State) | Chi-UNet | **Regression** | $1.63 \pm 0.79$ |
| | | **Flow** | $1.53 \pm 1.01$ |
| | Sudeep-DiT | **Regression** | $1.86 \pm 0.81$ |
| | | **Flow** | $1.34 \pm 0.97$ |
| | Chi-Transformer | **Regression** | $1.76 \pm 1.02$ |
| | | **Flow** | $1.40 \pm 0.99$ |

Table 17: Detailed: Per-architecture policy Lipschitz.

## F MULTI-MODALITY STUDY DETAILS

### F.1 Q FUNCTION ESTIMATION

To rule out the possibility of hidden multi-modality, we also plot Q functions for each action to see if there is any clear clustering pattern of $Q$ w.r.t. different actions in t-SNE visualization. Since we

only have access to expert actions rather than their policy, we estimate the Q function by Monte Carlo sampling with the learned flow policy. The detailed procedure is as follows:

Starting from one "critical state", we first sample $N$ actions

$$a^{(i)} = \Phi(o, z^{(i)}, s = 0, t = 1), \quad i = 1, \ldots, N, \quad z^{(i)} \sim \mathrm{N}(0, \mathbf{I}).$$

For each sampled action $a^{(i)}$, we execute one environment step to obtain the next observation $o'^{(i)}$ and immediate reward $r(o, a^{(i)})$. Then, starting from $o'^{(i)}$, we rollout the learned policy for $N_{\mathrm{MC}}$ episodes until termination (horizon $H$), and average the cumulative returns to obtain an estimate of the continuation value. Thus, the Q-value for action $a^{(i)}$ is approximated as:

$$Q_\Phi(a^{(i)}, o) = r(o, a^{(i)}) + \frac{1}{N_{\mathrm{MC}}} \sum_{j=1}^{N_{\mathrm{MC}}} \sum_{t=1}^{H} r\big(o_t^{(j)}, a_t^{(j)}\big). \tag{F.1}$$

We set the discount factor $\gamma = 1.0$ since rewards are sparse and triggered only at task completion. The reward for `Tool-Hang` and `Kitchen` is defined by the *final* success signal (with `Kitchen`'s success requiring all 4 subtasks to be completed). The reward for `Push-T` is defined by *final* coverage.

---

**Algorithm 2** Q Function Estimation via Monte Carlo Sampling

---

**Require:** Dataset $\mathcal{D}$, flow policy $\Phi$, reward function $r$, number of critical states $N_s{=}100$, number of action samples $N$, Monte Carlo samples $N_{\mathrm{MC}}$
**Ensure:** For each state $o$, pairs $\{(a^{(i)}, Q_\Phi(a^{(i)}, o))\}_{i=1}^{N}$
 1: $S \leftarrow$ identify $N_s$ critical states from $\mathcal{D}$        ▷ Select states with highest action ambiguity
 2: **for all** critical state $s \in S$ **do**
 3:     $o \leftarrow$ observation for state $s$
 4:     **for** $i = 1$ to $N$ **do**                ▷ Sample actions and compute Q estimates
 5:         $z^{(i)} \sim \mathrm{N}(0, \mathbf{I})$
 6:         $a^{(i)} \leftarrow \Phi(o, z^{(i)}, s{=}0, t{=}1)$
 7:         Execute $(o, a^{(i)})$ in env $\rightarrow$ obtain $o'^{(i)}$, $r^{(i)} = r(o, a^{(i)})$
 8:         **for** $j = 1$ to $N_{\mathrm{MC}}$ **do**                ▷ Monte Carlo rollouts from $o'^{(i)}$
 9:             Rollout $\Phi$ from $o'^{(i)}$ until horizon $H$ to get cumulative return $R_j^{(i)}$
10:         $Q_\Phi(a^{(i)}, o) \leftarrow r^{(i)} + \frac{1}{N_{\mathrm{MC}}} \sum_{j=1}^{N_{\mathrm{MC}}} R_j^{(i)}$
11:     Store $\{(a^{(i)}, Q_\Phi(a^{(i)}, o))\}_{i=1}^{N}$ for state $s$

---

The procedure above explicitly computes Q-values by rolling out trajectories separately for each sampled action.

### F.2 DETERMINISTIC DATASET GENERATION

To generate a deterministic dataset that completely eliminates any potential multi-modality, we follow a systematic process:

First, we train a flow expert policy $\Phi$ on the original dataset. Then, we collect a new dataset by rolling out this expert policy from different initial states (using different random seeds than those used during testing). Crucially, during rollout, we always evaluate the flow policy deterministically by setting the initial noise to zero: $z = 0$. This ensures that the policy produces deterministic actions given any observation, completely removing any stochasticity from the action generation process.

During data collection, we discard all failed trajectories to maintain the same success rate as the original dataset. We continue collecting until we reach the target number of trajectories $N_{\mathrm{traj}}$.

---

**Algorithm 3** Deterministic Dataset Generation

---

**Require:** Trained flow policy $\Phi$, target number of trajectories $N_{\text{traj}}$, maximum episode steps $T_{\text{max}}$
**Ensure:** Deterministic dataset $\mathcal{D}_{\text{det}}$
 1: $\mathcal{D}_{\text{det}} \leftarrow \emptyset$
 2: $n_{\text{collected}} \leftarrow 0$
 3: **while** $n_{\text{collected}} < N_{\text{traj}}$ **do**
 4:      Reset environment with new random seed
 5:      $o_0 \leftarrow$ initial observation
 6:      $\tau \leftarrow [(o_0, \cdot)]$                                 ▷ Initialize trajectory
 7:      **for** $t = 0$ to $T_{\text{max}} - 1$ **do**
 8:          $a_t \leftarrow \Phi(z = 0, o_t, s = 0, t = 1)$             ▷ Deterministic action
 9:          $o_{t+1}, r_t, \text{done} \leftarrow \text{env.step}(a_t)$
10:          $\tau \leftarrow \tau \cup [(o_t, a_t)]$
11:          **if** done **then**
12:             **break**
13:      **if** trajectory $\tau$ is successful **then**
14:          $\mathcal{D}_{\text{det}} \leftarrow \mathcal{D}_{\text{det}} \cup \{\tau\}$
15:          $n_{\text{collected}} \leftarrow n_{\text{collected}} + 1$
16: **return** $\mathcal{D}_{\text{det}}$

---

# G MANIFOLD ADHERENCE STUDY DETAILS

## G.1 VALIDATION LOSS IS NOT A GOOD PROXY FOR POLICY PERFORMANCE

To investigate whether validation loss serves as a reliable proxy for policy performance, we examine its relationship with success rates on `Tool-Hang` across different architectures given different training methods. Evidence that validation loss is poorly correlated with success rate can be seen by comparing flow policies with varying numbers of function evaluations (NFEs) and their corresponding validation losses. Table 18 demonstrates that increasing NFEs does not reduce validation loss, yet policy performance consistently improves. We hypothesize that higher NFEs introduce stronger inductive bias and regularization, which projects actions back onto the data manifold, thereby enhancing generalization.

| Architecture | Method | NFEs | Average Success Rate | $L_2$ Validation Loss |
|---|---|---|---|---|
| Chi-UNet | **Regression** | 1 | 0.54 | 0.063 |
| | **Flow** | 1 | 0.36 | 0.053 |
| | **Flow** | 3 | 0.44 | 0.052 |
| | **Flow** | 9 | 0.58 | 0.053 |
| Chi-Transformer | **Regression** | 1 | 0.18 | 0.084 |
| | **Flow** | 1 | 0.06 | 0.093 |
| | **Flow** | 3 | 0.72 | 0.092 |
| | **Flow** | 9 | 0.68 | 0.089 |
| Sudeep-DiT | **Regression** | 1 | 0.20 | 0.063 |
| | **Flow** | 1 | 0.62 | 0.082 |
| | **Flow** | 3 | 0.76 | 0.080 |
| | **Flow** | 9 | 0.76 | 0.080 |

**Table 18:** Comparison of validation loss and success rate across different architectures and methods on state-based `Tool-Hang`. The results show that validation loss is not a reliable proxy for policy performance.

## G.2 MANIFOLD ADHERENCE EVALUATION METHOD

To evaluate the manifold adherence, we compute the projection error of a predicted action $a$ onto the space spanned by expert actions at neighboring states. Concretely, given a state, we compute its $\ell_2$ distance to all states in the training set. Then, we pick $k$ nearest neighbor states and gather their

corresponding actions $A = [a^{(0)}, a^{(1)}, \dots, a^{(k)}]$. Lastly, we compute projection error by projecting $a$ to the column space of $A$: $\|a - P_A(a)\|_2 = \min_c \|a - Ac\|_2$.

## H    NEAREST NEIGHBOR HYPOTHESIS STUDY

Another popular hypothesis is that GCPs are learning a lookup table of observation-to-action mappings (Pari et al., 2021; He et al., 2025). This might be true for relatively simple tasks that do not require high precision and complex generalization, such as `Can`. However, for tasks that require higher precision and more contact, such as `Tool-Hang`, the nearest-neighbor/lookup-table assumption is insufficient to explain the success of GCPs. We evaluate the performance of a nearest-neighbor policy (VINN (Pari et al., 2021)) on state-based `Tool-Hang` and find that it achieves a success rate of only 12% as shown in Table 19. This is significantly lower than both flow and regression methods, indicating that the action manifold is not linearly spanned by the expert actions. Nevertheless, nearest-neighbor can still serve as a proxy for the expert action manifold, as it captures the general trend of actions—even though linear combinations of actions in the dataset cannot directly produce the correct action, the expert action manifold should not be too distant. Therefore, in this paper, we use nearest-neighbor as a proxy for the linearized expert action manifold rather than directly computing the distance between expert actions in the validation set and predicted actions.

| Action Chunk Size | Success Rate (%) |
|:---:|:---:|
| 1 | 0 |
| 8 | 4 |
| 16 | 12 |
| 32 | 2 |

**Table 19:** Performance of k-nearest neighbor policy on state-based `Tool-Hang` task. Using the same method as VINN with softmax over k=5 nearest neighbors.

### H.1    ACTION CHUNK SIZE STUDY

Another equivalent important factor is the action chunk size. Fig. 8 highlights the importance of action chunk size, where regression with larger action chunk can outperform flow with smaller action chunk size. Our ablation also indicates that **MIP** outperforms flow with smaller action chunk size and matches the performance of flow with larger action chunk size.

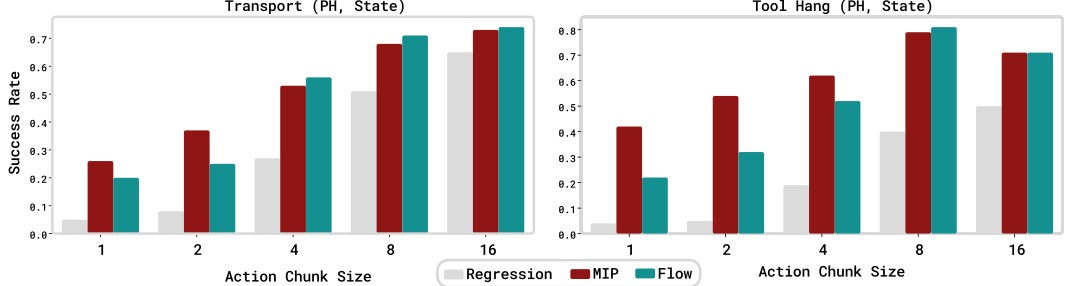

**Figure 8: Action chunk size ablation.** Success rate are averaged across 3 seeds, 3 architectures and 5 checkpoints on `Tool-Hang` and `Transport` tasks. Prediction horizon is set to powers of 2 to make sure it is compatible with `Chi-UNet` Architecture.

### H.2    LOSS NORM TYPE ABLATION STUDY

Previous work (Kim et al., 2024) shows that $\ell_1$ loss is superior to $\ell_2$ loss for regression policy. To test this hypothesis, we ablate the loss norm type and compare the performance of different methods with $\ell_1$ and $\ell_2$ loss. Table 20 shows the performance comparison of different methods with $\ell_1$ and $\ell_2$ loss, where we find that $\ell_1$ loss generally outperforms $\ell_2$ loss, especially for regression policy. However,

even with $\ell_1$ loss, we still observe that Regression $<$ **MIP** $\approx$ Flow, highlighting the importance of the stochasticity injection and iterative computation is independent of the loss function.

| Architecture | Method | Transport (ph) | Tool Hang (ph) |
|---|---|---|---|
| DiT | Regression L1 | 0.72/0.64 | 0.76/0.65 |
| DiT | Regression L2 | 0.50/0.45 | 0.50/0.37 |
| DiT | MIP L1 | 0.81/0.73 | 0.91/0.84 |
| DiT | MIP L2 | 0.79/0.69 | 0.92/0.85 |
| DiT | Flow L1 | 0.83/0.76 | 0.93/0.84 |
| DiT | Flow L2 | 0.81/0.71 | 0.89/0.75 |
| Transformer | Regression L1 | 0.67/0.57 | 0.44/0.33 |
| Transformer | Regression L2 | 0.65/0.54 | 0.31/0.25 |
| Transformer | MIP L1 | 0.80/0.68 | 0.85/0.77 |
| Transformer | MIP L2 | 0.80/0.69 | 0.80/0.72 |
| Transformer | Flow L1 | 0.77/0.69 | 0.81/0.71 |
| Transformer | Flow L2 | 0.79/0.65 | 0.73/0.61 |
| UNet | Regression L1 | 0.84/0.71 | 0.71/0.55 |
| UNet | Regression L2 | 0.66/0.59 | 0.73/0.59 |
| UNet | MIP L1 | 0.85/0.68 | 0.76/0.67 |
| UNet | MIP L2 | 0.81/0.72 | 0.82/0.71 |
| UNet | Flow L1 | 0.85/0.69 | 0.87/0.71 |
| UNet | Flow L2 | 0.83/0.75 | 0.87/0.73 |

**Table 20:** Comparison of L1 vs L2 norm across different methods and architectures. Report average/best performance across 5 checkpoints with 3 random seeds.

# I   DIVERSITY OF GCPS AND RCPS

A commonly believed hypothesis is that GCPs can express more diverse behaviors than RCPs by capturing the full distribution of expert actions (Shafiullah et al., 2022). We evelute different variants of GCPs and RCPs on FRANKA-KITCHEN, where the expert shows multiple task completion orders. As demonstrated in **??**, GCPs with both stochastic and deterministic sampling show similar task completion order diversity. Deterministic policies like regression and **MIP** (to be introduced in Section 4) also demonstrate similar task completion order diversity. This indicates that, given sparse expert demonstrations, both GCPs and RCP learns high-Lipschitz policies to switch between different modes given different observations (corresponding to (b.2) case in Figure 6). RCPs and GCPs are equally good at learning such behaviors (Figure 9) which explain why we see similar performance for both policy parametrizations, even on seemingly multi-modal tasks like FRANKA-KITCHEN.

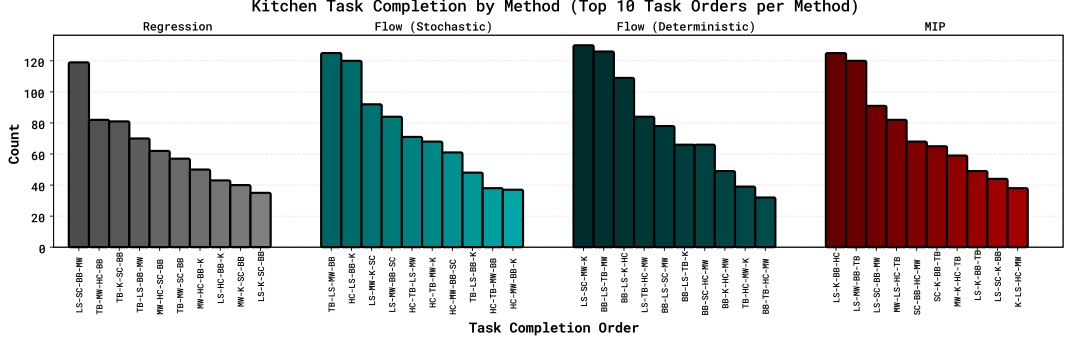

**Figure 9: Task completion order in Kitchen environment with different methods.** We plot the count of different task completion orders for different methods to evaluate the diversity of the policies. The x-axis shows the task completion order, where each sub-task is represented by its initials. For each run, we collect 1000 trajectories with the same seed shared by all methods. For flow, we evaluate both stochastic and deterministic modes.

## J   THEORETICAL ANALYSIS OF GCP'S EXPRESSIVITY

### J.1   FORMAL STATEMENT OF THEOREM 1

In this section, we introduce the notation and definition required for the subsequent proofs and provide the formal statement of Theorem 1 from the main text. Throughout, let $\| \cdot \|_\circ$ denote any matrix norm satisfying the property $\|X_1 X_2\|_\circ \le \|X_1\|_{\mathrm{op}} \|X_2\|_\circ$. In contrast to the notation used in the main text, we define $\Phi_{s,t}(a, o)$ as the solution at time $t$ of the ODE:

$$\frac{\mathrm{d}}{\mathrm{d}t} a_t = b_t^\star(a_t \mid o), \quad \text{with initial condition } a_s = a. \tag{J.1}$$

Note that $\Phi_{0,1}(a_0 = z, o)$ coincides with the definition of $\pi_\theta^\star(z, o)$ in the main text. Next, we define the notion of $\kappa$-log-concavity.

**Definition J.1** ($\kappa$-log-concavity). A distribution with density $\rho = e^{-V(x)}$ is said to be $\kappa$-log-concave if $V \in C^2(\mathbb{R}^d)$ and its Hessian satisfies $\nabla^2 V(x) \succcurlyeq \kappa \mathbf{I}$ for all $x \in \mathbb{R}^d$ and some $\kappa > 0$.

With this notation in place, we now state the formal version of Theorem 1.

**Theorem 2**. *Suppose that*

$$b_t^\star = \mathbb{E}[\dot{I}_t \mid I_t, o], \qquad \text{where } I_t = (1-t)a_0 + t a_1, \quad a_0 \sim \mathrm{N}(0, \mathbf{I}), \quad a_1 \sim \rho_1, \tag{J.2}$$

*where $\rho_1$ is $\kappa$-log-concave. Then, we have*

$$\|\nabla_o \Phi_{0,t}(a_0, o)\|_\circ \le \int_0^t \sqrt{\frac{\kappa(1-t)^2 + t^2}{\kappa(1-s)^2 + s^2}} \cdot \|\nabla_o b_s^\star(a_s \mid o)\|_\circ \mathrm{d}s. \tag{J.3}$$

*In particular, for $t = 1$ we obtain*

$$\|\nabla_o \Phi_{0,1}(a_0, o)\|_\circ \le \sqrt{1 + \kappa^{-1}} \int_0^1 \|\nabla_o b_s^\star(a_s \mid o)\|_\circ \mathrm{d}s. \tag{J.4}$$

**Remark J.1**. Theorem 1 follows immediately from the fact that both the operator and the Frobenius norms satisfy $\|X_1 X_2\|_\circ \le \|X_1\|_{\mathrm{op}} \|X_2\|_\circ$ together with the inequality Eq. (J.4).

### J.2   SUPPORTING LEMMAS

We state the supporting lemmas for proving Theorem 2 below and provide their proofs immediately for completeness. As a first step, we analyze the dynamical system satisfied by $\nabla_o \Phi_{s,t}(a, o)$.

**Lemma J.1**. *Define $a_t := \Phi_{0,t}(a_0, o)$ where $a_0$ is the initial condition, and define the matrices*

$$M_t := \nabla_o \Phi_{0,t}(a_0, o), \quad A_t := (\nabla_a b_t^\star)(a_t \mid o), \quad E_t := (\nabla_o b_t^\star)(a_t \mid o) \tag{J.5}$$

*Then,*

$$\frac{\mathrm{d}}{\mathrm{d}t} M_t = A_t M_t + E_t, \quad M_0 = 0 \tag{J.6}$$

*Proof.* Since $\Phi_{0,0}(a_0, o) = a_0$, $M_0 = 0$. Moreover,

$$\frac{\mathrm{d}}{\mathrm{d}t} \nabla_o \Phi_{0,t}(a_0, o) = \nabla_o \frac{\mathrm{d}}{\mathrm{d}t} \Phi_{0,t}(a_0, o) = \nabla_o (b_t^\star(\Phi_{0,t}(a_0, o) \mid o)) \tag{J.7}$$

$$= (\nabla_a b_t^\star)(a_t \mid o) \cdot \nabla_o \Phi_{0,t}(a_0, o) + (\nabla_o b_t^\star)(a_t \mid o) \tag{J.8}$$

$$\square$$

Note that, from the previous lemma, we may introduce $\Lambda_{s,t}$ as the solution to the matrix ODE

$$\frac{\mathrm{d}}{\mathrm{d}t} \Lambda_{s,t} = A_t \Lambda_{s,t}, \quad \Lambda_{s,s} = \mathbf{I}. \tag{J.9}$$

Moreover, it follows that

$$\Lambda_{s,t} = \nabla_a \Phi_{s,t}(a_s, o). \tag{J.10}$$

We are now ready to state the relation between $M_t$ and $\Lambda_{s,t}$.

**Lemma J.2.**

$$M_t = \int_0^t \Lambda_{s,t} E_s \mathrm{d}s. \tag{J.11}$$

*Proof.* Using $\frac{\mathrm{d}}{\mathrm{d}t}\Lambda_{0,t}^{-1} = -\Lambda_{0,t}^{-1}A_t$ and we consider the time derivative of $\Lambda_{0,t}^{-1}M_t$:

$$\frac{\mathrm{d}}{\mathrm{d}t}(\Lambda_{0,t}^{-1}M_t) = (\frac{\mathrm{d}}{\mathrm{d}t}\Lambda_{0,t}^{-1})M_t + \Lambda_{0,t}^{-1}(\frac{\mathrm{d}}{\mathrm{d}t}M_t) \tag{J.12}$$

$$= -\Lambda_{0,t}^{-1}A_t M_t + \Lambda_{0,t}^{-1}A_t M_t + \Lambda_{0,t}^{-1}E_t \tag{J.13}$$

$$= \Lambda_{0,t}^{-1}E_t. \tag{J.14}$$

Note that $\Lambda_{0,t}$ is invertible by uniqueness of the ODE solution in Eq. (J.9). Integrating both sides with respect to $t$ gives

$$\Lambda_{0,t}^{-1}M_t = \int_0^t \Lambda_{0,s}^{-1}E_s \mathrm{d}s. \tag{J.15}$$

Hence, we have

$$M_t = \Lambda_{0,t}\int_0^t \Lambda_{0,s}^{-1}E_s \mathrm{d}s. \tag{J.16}$$

Note that $\Lambda_{0,s}^{-1} = \Lambda_{s,0}$ and $\Lambda_{0,t}\cdot\Lambda_{s,0} = \Lambda_{s,t}$, we obtain

$$M_t = \int_0^t \Lambda_{s,t} E_s \mathrm{d}s. \tag{J.17}$$

$\square$

An immediate application of the triangle inequality and the property of $\|\cdot\|_\circ$ yields

$$\|M_t\|_\circ \le \int_0^t \|\Lambda_{s,t}\|_{\mathrm{op}}\|E_s\|_\circ \mathrm{d}s. \tag{J.18}$$

Moreover, $\|\Lambda_{s,t}\|_{\mathrm{op}}$ admits the bound:

**Lemma J.3.**

$$\|\Lambda_{s,t}\|_{\mathrm{op}} \le \exp\left(\int_s^t \|A_{s'}\|_{\mathrm{op}}\mathrm{d}s'\right). \tag{J.19}$$

*Proof.* Define $f_\omega(s,t) = \Lambda_{s,t}\omega$. We have

$$\frac{\mathrm{d}}{\mathrm{d}t}\|f_\omega(s,t)\|_2 = \frac{1}{\|f_\omega(s,t)\|_2}f_\omega(s,t)^\top\frac{\mathrm{d}}{\mathrm{d}t}f_\omega(s,t) \tag{J.20}$$

$$= \frac{1}{\|f_\omega(s,t)\|_2}\omega^\top\Lambda_{s,t}^\top A_t\Lambda_{s,t}\omega \tag{J.21}$$

$$\le \|A_t\|_{\mathrm{op}}\|f_\omega(s,t)\|_2. \tag{J.22}$$

By Gronwall's theorem and $\|f_\omega(s,s)\|_2 = \|\omega\|_2$, we obtain

$$\|f_\omega(s,t)\|_2 \le \|\omega\|_2\exp(\int_s^t \|A_{s'}\|_{\mathrm{op}}\mathrm{d}s'). \tag{J.23}$$

$\square$

To bound $\exp\left(\int_s^t \|A_{s'}\|_{\mathrm{op}}\mathrm{d}s'\right)$, we use the following result from (Daniels, 2025), included here for completeness.

**Theorem 3** (Restated; Theorem 6 in (Daniels, 2025)). *Suppose $\mu_0 \sim N(0, \mathbf{I})$ and $\mu_1$ is a $\kappa$-log-concave distribution with $\kappa > 0$. Define*

$$I_t = \alpha_t X_0 + \beta_t X_1, \qquad X_0 \sim \mu_0, \quad X_1 \sim \mu_1, \tag{J.24}$$

*and let $v_t(x)$ denote the corresponding flow field. Then,*

$$\nabla_x v_t(x) \preccurlyeq \frac{\kappa \alpha_t \dot{\alpha}_t + \beta_t \dot{\beta}_t}{\kappa \alpha_t^2 + \beta_t^2} \mathbf{I}. \tag{J.25}$$

With the result, we can bound $\exp\left(\int_s^t \|A_{s'}\|_{\mathrm{op}} \mathrm{d}s'\right)$ as follows.

**Lemma J.4.**

$$b_t^\star = \mathbb{E}[\dot{I}_t \mid I_t, o], \qquad \text{where } I_t = (1-t)a_0 + t a_1, \quad a_0 \sim N(0, \mathbf{I}), \quad a_1 \sim \rho_1, \tag{J.26}$$

*where $\rho_1$ is $\kappa$-log-concave. Then, we have*

$$\int_s^t \|\nabla_x b_{s'}^\star(a_{s'} \mid o)\|_{\mathrm{op}} \mathrm{d}s' \leq \log \sqrt{\frac{\kappa(1-t)^2 + t^2}{\kappa(1-s)^2 + s^2}} \tag{J.27}$$

*Proof.* By leveraging Theorem 3 for each condition $o$, we have

$$\nabla_a b_{s'}^\star(a_{s'} \mid o) \preccurlyeq \frac{\kappa \alpha_{s'} \dot{\alpha}_{s'} + \beta_{s'} \dot{\beta}_{s'}}{\kappa \alpha_{s'}^2 + \beta_{s'}^2} \mathbf{I}, \tag{J.28}$$

then we have

$$\|\nabla_a b_{s'}^\star(a_{s'} \mid o)\|_{\mathrm{op}} \leq \frac{\kappa \alpha_{s'} \dot{\alpha}_{s'} + \beta_{s'} \dot{\beta}_{s'}}{\kappa \alpha_{s'}^2 + \beta_{s'}^2}. \tag{J.29}$$

Integrating both sides, we obtain

$$\int_s^t \|\nabla_a b_{s'}^\star(a_{s'} \mid o)\|_{\mathrm{op}} \mathrm{d}s' \leq \int_s^t \frac{\kappa \alpha_{s'} \dot{\alpha}_{s'} + \beta_{s'} \dot{\beta}_s'}{\kappa \alpha_{s'}^2 + \beta_{s'}^2} \mathrm{d}s' \tag{J.30}$$

$$= \frac{1}{2} \log(\kappa \alpha_{s'}^2 + \beta_{s'}^2) \Big|_s^t \tag{J.31}$$

$$= \log \sqrt{\frac{\kappa \alpha_t^2 + \beta_t^2}{\kappa \alpha_s^2 + \beta_s^2}}. \tag{J.32}$$

By substitute $\alpha_t = 1 - t$ and $\beta_t = t$, we have

$$\int_s^t \|\nabla_a b_{s'}^\star(a_{s'} \mid o)\|_{\mathrm{op}} \mathrm{d}s' \leq \log \sqrt{\frac{\kappa(1-t)^2 + t^2}{\kappa(1-s)^2 + s^2}}. \tag{J.33}$$

$\square$

With the preceding components in place, we now establish Theorem 2.

## J.3 PROOF OF THEOREM 2

By combining Eq. (J.18), Lemma J.3, and Lemma J.4, we have

$$\|\nabla_o \Phi_{0,t}(a_0, o)\|_\circ \leq \int_0^t \sqrt{\frac{\kappa(1-t)^2 + t^2}{\kappa(1-s)^2 + s^2}} \cdot \|\nabla_o b_s^\star(a_s \mid o)\|_\circ \mathrm{d}s. \tag{J.34}$$

For $t = 1$, the function $s \mapsto \kappa(1-s)^2 + s^2$ attains its minimum at $s = \frac{\kappa}{\kappa+1}$. Applying Holder's inequality then yields

$$\|\nabla_o \Phi_{0,1}(a_0, o)\|_\circ \leq \int_0^1 \sqrt{\frac{1}{\kappa(1-s)^2 + s^2}} \cdot \|\nabla_o b_s^\star(a_s \mid o)\|_\circ \mathrm{d}s \tag{J.35}$$

$$\leq \max_{s \in [0,1]} \left(\sqrt{\frac{1}{\kappa(1-s)^2 + s^2}}\right) \cdot \int_0^1 \|\nabla_o b_s^\star(a_s \mid o)\|_\circ \mathrm{d}s \tag{J.36}$$

$$= \sqrt{1 + \kappa^{-1}} \int_0^1 \|\nabla_o b_s^\star(a_s \mid o)\|_\circ \mathrm{d}s. \tag{J.37}$$

## K  REGULARIZATION DOES NOT ACCOUNT FOR MANIFOLD ADHERENCE

In this section, we analyze a linear, population-level surrogate for **MIP** to test whether *implicit regularization*, instantiated via ridge regression and a two-step **MIP**-like iteration, can explain the observed manifold adherence. We mimic the two passes of **MIP** with two ridge-regularized linear regressions: (i) a regression where the ridge penalty is applied to the observation-to-action map, and (ii) a regression where the ridge penalty is applied to the action-to-action map. We then compose the two fitted maps to obtain the two-stage inference used by **MIP**. As shown below, it instead yields smooth spectral shrinkage and does not make the manifold absorbing.

Throughout we work in expectation (population covariances), so that conclusions reflect model structure rather than finite-sample effects. We assume independence of $o$, $z$, and the additive noise $\eta$, and that the inverses we write exist (otherwise interpret as pseudoinverses on the relevant supports).

**Setup.** Observations $o \in \mathbb{R}^d$ and actions $a \in \mathbb{R}^d$ follow the linear model

$$a = \Theta^* o + \eta, \qquad \eta \sim \mathrm{N}(0, \Sigma_\eta = \eta^2 \mathbf{I}),$$

with $\eta \perp o$. Let $z \sim \mathrm{N}(0, \Sigma_z)$ be an auxiliary signal, independent of $(o, a, \eta)$, and define $w := c_1 a + c_2 z$. We consider linear predictors $\hat{a} = Bo + Cw$ with ridge regularization applied either to $B$ (Section K.1) or to $C$ (Section K.2). This reflects the use of both the observation $o$, and the action $a$, in prediction.

### K.1  RIDGE REGRESSION FOR OBSERVATION-TO-ACTION MAPPING (PENALTY ON $B$)

We solve

$$\min_{B,C} \mathbb{E}\|Bo + Cw - a\|^2 + \lambda\|B\|_F^2.$$

Define

$$X := \begin{bmatrix} o \\ w \end{bmatrix}, \qquad \Psi := \begin{bmatrix} B & C \end{bmatrix},$$

and, at the population level, the second-moment blocks

$$\Sigma_{11} := \mathbb{E}[o^{\otimes 2}] = \Sigma_o,$$
$$\Sigma_{12} := \mathbb{E}[ow^\top] = \mathbb{E}[o(c_1 a + c_2 z)^\top] = c_1 \Sigma_o \Theta^{*\top},$$
$$\Sigma_{21} := \Sigma_{12}^\top,$$
$$\Sigma_{22} := \mathbb{E}[w^{\otimes 2}] = c_1^2(\Theta^* \Sigma_o \Theta^{*\top} + \Sigma_\eta) + c_2^2 \Sigma_z.$$
$$\Sigma_{a1} := \mathbb{E}[ao^\top] = \Theta^* \Sigma_o,$$
$$\Sigma_{a2} := \mathbb{E}[aw^\top] = c_1(\Theta^* \Sigma_o \Theta^{*\top} + \Sigma_\eta),$$

where, for any vector $x$, we write $x^{\otimes 2} := xx^\top$.

The objective can be written in trace form as

$$\mathcal{L} = \mathbb{E}[(a - \Psi X)^\top(a - \Psi X)] + \lambda\|B\|_F^2$$
$$= \mathrm{tr}(\mathbb{E}[(a - \Psi X)(a - \Psi X)^\top]) + \lambda\mathrm{tr}(BB^\top)$$

Dropping terms that are constant in $(B, C)$, let

$$\Sigma_X := \mathbb{E}[XX^\top] = \begin{bmatrix} \Sigma_{11} & \Sigma_{12} \\ \Sigma_{21} & \Sigma_{22} \end{bmatrix}, \qquad \Sigma_{aX} := \mathbb{E}[aX^\top] = \begin{bmatrix} \Sigma_{a1} & \Sigma_{a2} \end{bmatrix}.$$

Then

$$\mathcal{L} = -2\operatorname{tr}(\Psi\Sigma_{aX}^\top) + \operatorname{tr}(\Psi\Sigma_X\Psi^\top) + \lambda\operatorname{tr}(BB^\top).$$

Differentiating gives

$$\nabla_B\mathcal{L} = -2\Sigma_{a1} + 2B\Sigma_{11} + 2C\Sigma_{21} + \lambda 2B,$$
$$\nabla_C\mathcal{L} = -2\Sigma_{a2} + 2B\Sigma_{12} + 2C\Sigma_{22}.$$

Setting the gradients to zero yields the normal equations:

$$B(\Sigma_{11} + \lambda\mathbf{I}) + C\Sigma_{21} = \Sigma_{a1}$$
$$B\Sigma_{12} + C\Sigma_{22} = \Sigma_{a2}.$$

Solving the linear system (e.g., by block elimination) yields

$$B = \underbrace{(\Sigma_{a1}\Sigma_{21}^{-1}\Sigma_{22} - \Sigma_{a2})}_{\text{(i)}}\underbrace{[(\Sigma_{11} + \lambda\mathbf{I})\Sigma_{21}^{-1}\Sigma_{22} - \Sigma_{12}]^{-1}}_{\text{(ii)}},$$
$$C = [\Sigma_{a1} - B(\Sigma_{11} + \lambda\mathbf{I})]\Sigma_{21}^{-1}.$$

Using $\Sigma_{21} = c_1\Theta^*\Sigma_o$, we have $\Sigma_{21}^{-1} = \frac{1}{c_1}(\Theta^*\Sigma_o)^{-1}$. For (i),

$$\begin{aligned}
\text{(i)} &= \Sigma_{a1}\Sigma_{21}^{-1}\Sigma_{22} - \Sigma_{a2} \\
&= \Theta^*\Sigma_o\frac{1}{c_1}(\Theta^*\Sigma_o)^{-1}[c_1^2(\Theta^*\Sigma_o\Theta^{*\top} + \Sigma_\eta) + c_2^2\Sigma_z] - c_1(\Theta^*\Sigma_o\Theta^{*\top} + \Sigma_\eta) \\
&= c_1(\Theta^*\Sigma_o\Theta^{*\top} + \Sigma_\eta) + \frac{c_2^2}{c_1}\Sigma_z - c_1(\Theta^*\Sigma_o\Theta^{*\top} + \Sigma_\eta) \\
&= \frac{c_2^2}{c_1}\Sigma_z.
\end{aligned}$$

For (ii),

$$\begin{aligned}
\text{(ii)} &= [(\Sigma_{11} + \lambda\mathbf{I})\Sigma_{21}^{-1}\Sigma_{22} - \Sigma_{12}]^{-1} \\
&= [(\Sigma_o + \lambda\mathbf{I})\frac{1}{c_1}(\Theta^*\Sigma_o)^{-1}[c_1^2(\Theta^*\Sigma_o\Theta^{*\top} + \Sigma_\eta) + c_2^2\Sigma_z] - c_1\Sigma_o(\Theta^*)^\top]^{-1} \\
&= [(\Sigma_o + \lambda\mathbf{I})[c_1\Theta^{*\top} + c_1(\Theta^*\Sigma_o)^{-1}\Sigma_\eta + \frac{c_2^2}{c_1}(\Theta^*\Sigma_o)^{-1}\Sigma_z] - c_1\Sigma_o(\Theta^*)^\top]^{-1} \\
&= [\lambda c_1(\Theta^*)^\top + (\Sigma_o + \lambda\mathbf{I})(c_1(\Theta^*\Sigma_o)^{-1}\Sigma_\eta + \frac{c_2^2}{c_1}(\Theta^*\Sigma_o)^{-1}\Sigma_z)]^{-1}
\end{aligned}$$

**Isotropic specialization.** Take $\Sigma_o = \mathbf{I}$, $\Sigma_\eta = \eta^2\mathbf{I}$, $\Sigma_z = \mathbf{I}$, and $\Theta^* = \operatorname{diag}(s_i)$. Then

$$\text{(i)} = \frac{c_2^2}{c_1}\mathbf{I},$$

$$\begin{aligned}
\text{(ii)} &= [\lambda c_1(\Theta^*)^\top + (1 + \lambda)c_1\eta^2(\Theta^*)^{-1} + (1 + \lambda)\frac{c_2^2}{c_1}(\Theta^*)^{-1}]^{-1} \\
&= \operatorname{diag}\left([\lambda c_1 s_i + \frac{(1 + \lambda)(c_1\eta^2 + \frac{c_2^2}{c_1})}{s_i}]^{-1}\right) \\
&= \operatorname{diag}\left(\frac{c_1 s_i}{\lambda c_1^2 s_i^2 + (1 + \lambda)(c_1^2\eta^2 + c_2^2)}\right)
\end{aligned}$$

Hence, we obtain

$$B = \text{diag}\left(\frac{c_2^2 s_i}{\lambda c_1^2 s_i^2 + (1+\lambda)(c_1^2\eta^2 + c_2^2)}\right), \quad C = \text{diag}\left(\frac{1}{c_1}\left(1 - \frac{(1+\lambda)c_2^2}{\lambda c_1^2 s_i^2 + (1+\lambda)(c_1^2\eta^2 + c_2^2)}\right)\right).$$

Define the shrinkage factor for the $i$-th singular direction by $\rho_i := B_{ii}/s_i$. Then,

$$\rho_i = \frac{c_2^2}{\lambda c_1^2 s_i^2 + (1+\lambda)(c_1^2\eta^2 + c_2^2)}. \tag{K.1}$$

**Key implications:**

- **Ridge on $B$ ($\lambda > 0$) makes shrinkage $s_i$-dependent.** From (K.1), the factor decreases with $s_i$ (because the denominator has $\lambda c_1^2 s_i^2$). So **larger** singular directions are shrunk **more** when $\lambda > 0$.

- **If no ridge ($\lambda = 0$).** $B = \text{diag}(\frac{c_2^2 s_i}{c_1^2\eta^2 + c_2^2})$: shrinkage is constant across $i$.

- **If no ridge and no noise ($\lambda = 0, \eta = 0$).** $B = \Theta^*$—no shrinkage (recovers the standard solution).

- **If no auxiliary $z$-signal ($c_2 = 0$).** $B = 0$.

### K.2  RIDGE REGRESSION FOR ACTION-TO-ACTION MAPPING (PENALTY ON $C$)

We now solve

$$\min_{B,C}; \mathbb{E}\|Bo + Cw - a\|^2 + \lambda\|C\|_F^2.$$

The normal equations become

$$B\Sigma_{11} + C\Sigma_{21} = \Sigma_{a1}$$
$$B\Sigma_{12} + C(\Sigma_{22} + \lambda\mathbf{I}) = \Sigma_{a2}.$$

Solving gives the closed-form estimators:

$$B = (\Sigma_{a1} - C\Sigma_{21})\Sigma_{11}^{-1},$$
$$C = \underbrace{(\Sigma_{a2} - \Sigma_{a1}\Sigma_{11}^{-1}\Sigma_{12})}_{(i)}\underbrace{(\Sigma_{22} + \lambda\mathbf{I} - \Sigma_{21}\Sigma_{11}^{-1}\Sigma_{12})^{-1}}_{(ii)}.$$

In the isotropic specialization $\Sigma_o = \mathbf{I}$, $\Sigma\eta = \eta^2\mathbf{I}$, $\Sigma_z = \mathbf{I}$, $\Theta^* = \text{diag}(s_i)$, one obtains

$$B = \text{diag}\left(\frac{(c_2^2 + \lambda)s_i}{c_1^2\eta^2 + c_2^2 + \lambda}\right), \qquad C = \frac{c_1\eta^2}{c_1^2\eta^2 + c_2^2 + \lambda}\mathbf{I}.$$

### K.3  TWO-PASS LINEAR SURROGATE OF **MIP**

Let $(B_1, C_1)$ denote the solution of Section K.1 with ridge $\lambda_1$ on $B$, and $(B_2, C_2)$ denote the solution of §K.2 with ridge $\lambda_2$ on $C$. To mimic the **MIP** two-pass inference rule in (4.3), we consider

$$\hat{a}_0 \leftarrow B_1 o, \qquad \hat{a} \leftarrow B_2 o + c_1 C_2 \hat{a}_0.$$

We obtain

$$\hat{a} \leftarrow \underbrace{(B_2 + c_1 C_2 B_1)}_{=:\hat{\Phi}} o.$$

Note that $c_1$ serves as the analogue of $t_\star$ from the main text. From Sections K.1 and K.2 we then obtain

$$\begin{aligned}
\hat{\Phi} &= (B_2 + c_1 C_2 B_1) \\
&= \text{diag}\left(\frac{(c_2^2 + \lambda_2)s_i}{c_1^2\eta^2 + c_2^2 + \lambda_2} + c_1\frac{c_1\eta^2}{c_1^2\eta^2 + c_2^2 + \lambda_2}\frac{c_2^2 s_i}{\lambda_1 c_1^2 s_i^2 + (1+\lambda_1)(c_1^2\eta^2 + c_2^2)}\right) \\
&= \text{diag}\left(\frac{s_i}{c_1^2\eta^2 + c_2^2 + \lambda_2}\left[c_2^2 + \lambda_2 + \frac{c_1^2\eta^2 c_2^2}{\lambda_1 c_1^2 s_i^2 + (1+\lambda_1)(c_1^2\eta^2 + c_2^2)}\right]\right).
\end{aligned} \tag{K.2}$$

Moreover, the shrink factor will be

$$\frac{\hat{\Phi}_{ii}}{s_i} = \frac{1}{c_1^2\eta^2 + c_2^2 + \lambda_2}\left[c_2^2 + \lambda_2 + \frac{c_1^2\eta^2 c_2^2}{\lambda_1 c_1^2 s_i^2 + (1+\lambda_1)(c_1^2\eta^2 + c_2^2)}\right]. \tag{K.3}$$

**Key implications:**

- **If no ridge on B ($\lambda_1 = 0$).** No $s_i$-dependent shrinkage.

- **If no noise ($\eta = 0$).** Just same as the regular case: $\hat{\Phi} = \Theta^*$.

- **Signal-to-noise effect.** The quantity in (K.2) rises with $s_i$ and falls with $\eta$, mildly favoring signal over noise by damping noisy directions.

**Why this composition is a plausible proxy.** The first stage applies a ridge penalty to the observation-to-action parameters $B$ and predicts an interpolant action from observations alone, as in the $t = 0, z = 0$ pass of **MIP**. We use ridge here as a canonical proxy for implicit regularization in the linear setting. The first stage applies a ridge penalty to the action-to-action parameters $C$. It takes the interpolant action input (near $t = 1$) together with $o$ and produces the final output. Composing the two yields the operator $\hat{\Phi}$ in (K.2), which is the linear analogue of the two-pass prediction of (4.3).

**Why shrinkage does not yield manifold adherence.** The operator in (K.2) acts as a spectral shrinker: because the factors in (K.3) decrease with $s_i$ (for $\lambda_1 > 0$), it attenuates the dominant directions more than the weak ones—contrary to a projection onto a manifold, which would preserve principal directions and damp small and noisy modes. Since these factors lie in $(0, 1]$ and vary smoothly with $s_i$, $\eta$, and $\lambda_1, \lambda_2$, the map lacks any projection-like behavior: once a point is off-manifold, it is neither returned to nor retained on any low-dimensional subspace. Thus, implicit regularization alone, even with the two-pass composition of **MIP**, cannot account for the observed manifold adherence.

## L  TOY EXPERIMENTS: TESTING THE FUNCTION APPROXIMATION CAPABILITIES OF REGRESSION AND FLOW MODELS

### L.1  OVERVIEW

This appendix summarizes an empirical comparison of training paradigms (regression, flow matching, straight flow, MIP) for function approximation with geometric constraints across three tasks: scalar reconstruction, high-dimensional projection with subspace constraints, and Lie algebra rotations. Experiments operate in low-data regimes (50 training samples) using concatenation and FiLM architectures, with results averaged across multiple random seeds.

### L.2  EVALUATION METRICS

**Reconstruction:** L1 and L2 errors measure point-wise approximation quality between predictions $\hat{f}(c)$ and targets $f(c)$.

**Projection:** Three metrics assess geometric constraints in piecewise-constant projection structure: *subspace diagonal* quantifies predictions outside correct subspace $P_i$ for interval $i$, *off-diagonal* tests cross-interval generalization with mismatched projections, and *boundary* measures smoothness at interval transitions using combined adjacent subspace projections. All metrics use normalized form $\|(I - P)\hat{f}\|/\|\hat{f}\|$.

**Lie Algebra:** *Cosine similarity* measures angular alignment between predicted and true rotation directions. *Projection metric* quantifies normalized perpendicular error relative to the rotation axis span.

## L.3    KEY FINDINGS

**Task-Dependent Performance:** Regression-based approaches achieve lowest L2 reconstruction error ($0.003197 \pm 0.000525$ with L2 loss and FiLM), consistently outperforming flow-based methods on point-wise approximation tasks.

**Flow Methods Excel at Projections:** Flow-based training demonstrates superior geometric constraint satisfaction. Straight flow (flow matching without time conditioning) achieves best boundary projection ($0.009769 \pm 0.001630$) and Lie algebra projection metrics ($0.063612 \pm 0.000952$), indicating beneficial geometric biases from learning probability transport.

**MIP Competitive Performance:** MIP combines direct regression with denoising regularization, achieving near-optimal reconstruction while maintaining reasonable geometric constraint satisfaction across tasks.

## L.4    TRAINING LOSS CONSIDERATIONS

Results focus on L2-trained models, providing mathematically grounded objectives for both regression and flow paradigms. While alternative loss functions were evaluated empirically, flow-based L1 training lacks principled derivation as conditional flow matching is naturally defined for squared error.

## L.5    ARCHITECTURAL OBSERVATIONS

Both concatenation and FiLM architectures demonstrated competitive performance with no consistent dominance. FiLM showed marginal advantages on certain geometric metrics for flow-based methods, suggesting affine feature modulation may better capture conditional dependencies in probability transport.

## L.6    IMPLICATIONS FOR METHOD SELECTION

- Tasks prioritizing point-wise reconstruction: regression-based training with L2 loss offers superior accuracy and computational efficiency.
- Tasks requiring geometric constraint satisfaction: flow-based training provides significant advantages despite increased evaluation cost.
- Straight flow's success suggests time conditioning may be unnecessary, enabling simpler models with competitive performance.

## L.7    EXPERIMENTAL DETAILS

Study encompasses 540 runs: 5 modes (regression, flow, straight flow, MIP, MIP one-step) $\times$ 2 losses $\times$ 2 architectures $\times$ 3 tasks $\times$ 3 seeds. Configuration: 256 hidden dimensions, 3 layers, ReLU, batch size 32, 50k epochs, Adam with lr=0.001. Evaluation: 100k test samples; flow methods use Euler integration with 9 ODE steps.

**Full Report:** `https://github.com/simchowitzlabpublic/much-ado-about-noising/blob/main/toyexp/toyexp/report/report.pdf` **Code:** `https://github.com/simchowitzlabpublic/much-ado-about-noising/tree/main/toyexp/toyexp`

# M APPENDIX FOR SECTION 2

## M.1 MARKOV DECISION PROCESSES CONFIGURATION

We consider a Markov Decision Process $\mathcal{M} = (\mathcal{S}, \mathcal{A}, R, P, P_0)^2$ with the state space $\mathcal{S}$, the action space $\mathcal{A}$, the reward $R(s, a)^3$ obtained by taking action $a$ in state $s$, the transition dynamics $P : \mathcal{S} \times \mathcal{A} \to \Delta(\mathcal{S})$, and the initial-state distribution $P_0 \in \Delta(\mathcal{S})$. To formulate the success rate (i.e., performance) in this setting, we define the reward function as:

$$R(s, a) = \begin{cases} 1, & \text{if the task is successful under } (s, a), \\ 0, & \text{otherwise.} \end{cases} \tag{M.1}$$

Under this definition of rewards, the expected return of a policy $\pi$ is $J(\pi) = \mathbb{E}[\sum_t R(s_t, a_t)]$, which reduces to $\mathbb{P}[\text{success under } \pi]$. Hence, $J(\pi)$ exactly equals the success rate of policy $\pi$.

## M.2 INTEGRATED FLOW PREDICTION

For completeness, we provide the flow ODE as

$$\frac{\mathrm{d}}{\mathrm{d}t} a_t = b_t(a_t \mid o) \qquad \text{starting from} \qquad a_0 = z. \tag{M.2}$$

The associated integrated flow prediction is given by

$$\Phi_\theta(z \mid o) = z + \int_0^1 b_t(a_t \mid o)\mathrm{d}t. \tag{M.3}$$

In practice, to approximate the ODE solution for sampling, we employ the following discretized Euler integration.

**Definition M.1** (Discretized Euler Integration). We discretize the time interval $[0, 1]$ to $N$ steps with step size $h = 1/N$. The iterates are then updated according to

$$a_{k+1} = a_k + h\, b_{hk}(a_k \mid o), \quad k = 0, 1, \ldots, N - 1. \tag{M.4}$$

The final iterate $a_N$ serves as the Euler approximation $\Phi_{\theta,\mathrm{eul}}(z \mid o)$. We also refer to $N$ as the *Number of Function Evaluations (NFEs)*.

# N LLM USAGE

We used LLMs only for minor language polishing (grammar and wording) and to assist with literature search. All technical content, experiments, and conclusions were created and verified by the authors.

---

[2]For simplicity, we consider the MDP case in this context by identifying the state with the observation defined in 2. More generally, one may consider a Partially Observable Markov Decision Process (POMDP), where the agent receives observation $o$ emitted by an underlying latent state $s$.

[3]For ease of exposition, we use the same notation for rewards defined on random variables and their distributions.

