# OpenReview forum: "Much Ado About Noising: Dispelling the Myths of Generative Robotic Control"
_ICLR.cc/2026/Conference — ICLR 2026 Poster_

### Official Review · Reviewer_zgJS · 2025-10-26

**Soundness:** 3
**Presentation:** 3
**Contribution:** 3
**Rating:** 8
**Confidence:** 4

**Summary:**

The paper questions why generative policies, specifically diffusion-based and flow-based imitation learning models, appear to outperform regression-based policies in robot learning. It argues that their success is not due to better distribution modeling or multimodality capture, but rather to algorithmic structure. The paper examined a few hypotheses and then concluded that GCPs are not fundamentally superior distribution learners; their strength arises from how their training and inference processes structure learning.

**Strengths:**

- The paper conducted comprehensive experiments and had careful comparison with architecture parity and identical backbones, ensuring fairness and validating the hypothesis. It has a large-scale benchmark study across 27 imitation learning datasets that isolates the actual causes of GCP success.
- It separates generative modeling goals from control objectives and proposes a new non-generative baseline that rivals flow policies MIP.
- It shows that distribution fitting is largely irrelevant for single-task control, which is a major conceptual correction.
- The insight that a two-step regression model can rival state-of-the-art flow policies is surprising.

**Weaknesses:**

- The experiments fix inference to deterministic modes, although guided sampling or diffusion-based control might behave differently.
- The study focuses on a single-task setting, while multi-task or goal-conditioned GCPs might still benefit from distributional modeling.
- The paper does not test whether C2+C3 advantages persist under RL fine-tuning or long-horizon planning, which is crucial for robotic applications.

**Questions:**

- The paper empirically shows that the combination of stochasticity and iteration works. Is the improvement due to smoother gradient landscapes, implicit ensemble effects, or noise-induced curvature exploration or is there any intuition?
- Can MIP generalize across different goals or objects without retraining? This would test whether its simplicity generalizes or overfits. Compared to it, flow-based policies often work well in multi-task settings where distributional conditioning is crucial.
- How does MIP perform under camera noise, lighting changes, or slight object translation? Does MIP maintain robustness to sensor noise or domain shift? Since MIP is deterministic, it may degrade fast.

---

> ### Author Response · Authors · 2025-11-19
> **Response to Reviewer zgJS**
>
> We thank the reviewer for their insightful comments. We have conducted additional experiments on multi-task learning (LIBERO), stochastic integrators, and robustness to address your concerns.
>
> ## Scope: Single-Task vs Multi-Task
> We acknowledge the community shift toward multi-task learning. However, our single-task benchmarks were selected specifically because the original claims of GCP superiority over regression were drawn from these settings [1-3]. Our goal was to rigorously test those mechanisms.
>
> To address the generalizability concern, we fine-tuned PI0 on the **LIBERO benchmark**, which requires a single policy to solve 130 tasks. As shown below, MIP achieves performance comparable to (or better than) Flow with significantly fewer function evaluations (NFE). Notably, simple Regression also performs well, reinforcing our conclusion that capturing multi-modal distributions is not the primary driver of success, even in multi-task regimes.
>
> | Task            | Libero Object | Libero Goal | Libero Spatial | Libero 10 |
> | :-- | :----- | :--- | :---- | :----- |
> | Regression      | 92.6          | 94.6        | 97.2           | 78.0      |
> | Flow (NFE=10)   | **97.4**      | 95.0        | 95.8           | 81.6      |
> | **MIP (NFE=2)** | 95.8          | **95.2**    | **97.6**       | **82.2**  |
>
> **Table: LIBERO Benchmark Results.** Comparison using `lerobot`’s pi0 fine-tuning code (50 prediction steps, 5 action steps). MIP ($t^*=0.9$) matches Flow performance with 5x fewer inference steps.
>
> ## Deterministic Inference vs Guided Sampling
> **Clarification on Evidence C:** "Deterministic" in this context refers to the *data collection policy* (a flow-from-zero expert), not the inference mode. This result demonstrates that GCPs outperform RCPs even when the training data is fully deterministic, ruling out the hypothesis that GCP superiority stems purely from modeling multi-modal data.
>
> **Stochastic Inference:** While we use deterministic ODE integrators (standard in [1-3]), we evaluated an SDE integrator (similar to DDPM) per your suggestion. As shown below, SDE offers only marginal gains (+0.01-0.05) but requires ~50x more steps.
>
> | Integrator | ODE (NFE=9) | SDE (NFE=500) |
> | :--------- | :------- | :--- |
> | Transport  | 0.77        | 0.78          |
> | Tool Hang  | 0.75        | 0.80          |
>
> **Table: Integrator Comparison.** Average success on our most challenging tasks (3 seeds, 3 architectures).
>
> Regarding guided sampling, we agree it is promising for planning [4], but view it as future work for this behavior cloning focus.
>
> ## Mechanism Behind C2 + C3
> Empirically, C2 (Noise Injection) and C3 (Iterative Denoising) improve Out-Of-Distribution (OOD) behavior and manifold adherence.
>
> Theoretically, we find that standard implicit regularization arguments do not fully explain this. In our revision, we will add additional results to show that a "two-step training, single-step inference" baseline offers little benefit over regression, suggesting the gain is not solely due to optimization properties. Intuitively, the benefit stems from the strong inductive bias provided by the decoupled two-step training combined with noise injection, which guides the policy to produce plausible actions on the data manifold.
>
> ## Generalization Across Goals / Objects
> We test generalizability via rigorous environment randomization in all experiments. Furthermore, the Kitchen benchmark used in the main text is inherently multi-task (4 tasks, arbitrary order).
> Most importantly, our new **LIBERO results** (see Table in Section 1) demonstrate that MIP generalizes effectively across 130 distinct tasks and objects/goals without retraining, matching Flow-based policies. This indicates that explicit distributional modeling is not a prerequisite for multi-task success.
>
> ## Robustness to Observation Noise / Domain Shift
> We conducted a robustness test on the vision-based "Tool Hang" task by injecting Gaussian noise into the input images. Both MIP and Flow maintain performance at moderate noise levels and degrade significantly at $\sigma > 0.075$. We observe no significant robustness gap between the methods.
>
> | Noise Std ($\sigma$) | 0.0  | 0.025 | 0.05 | 0.075 | 0.1  |
> | :------------------- | :--- | :---- | :--- | :---- | :--- |
> | Regression           | 0.30 | 0.24  | 0.20 | 0.08  | 0.00 |
> | MIP                  | 0.80 | 0.72  | 0.60 | 0.05  | 0.00 |
> | Flow                 | 0.75 | 0.65  | 0.50 | 0.10  | 0.00 |
>
> **Table: Robustness to Image Noise.** Average success over 50 trials (3 seeds).
>
> ---
>
> [1] Chi et al., "Diffusion Policy: Visuomotor Policy Learning via Action Diffusion", RSS 2023.
>
> [2] Ze et al., "3D Diffusion Policy: Generalizable Visuomotor Policy Learning via Simple 3D Representations", RSS 2024.
>
> [3] Zhang et al., "FlowPolicy: Enabling Fast and Robust 3D Flow-Based Policy via Consistency Flow Matching for Robot Manipulation", AAAI 2025.
>
> [4] Janner et al., "Planning with Diffusion for Flexible Behavior Synthesis", ICML 2022.

---

> > ### Comment · Reviewer_zgJS · 2025-11-27
> > **Response to Submission21337 Authors**
> >
> > Thank the authors for the detailed response. Especially for the additional experiments and your conclusion looks promising. I have no major concerns and will keep my positive recommendation. I would love to discuss with the other reviewers.

---

### Official Review · Reviewer_Kdio · 2025-10-29

**Soundness:** 3
**Presentation:** 3
**Contribution:** 3
**Rating:** 6
**Confidence:** 3

**Summary:**

This paper investigates why generative control policies (GCPs)—such as flow- or diffusion-based action models—perform better than standard regression policies in behavior cloning (BC).
Rather than assuming the benefit arises from distributional modeling (learning multimodal
𝑝
(
𝑎
∣
𝑜
)
p(a∣o)), the authors dissect GCPs into three conceptual components:

C1: Distributional learning (learning a stochastic action distribution)

C2: Stochastic training (injecting noise during training)

C3: Supervised iterative computation (multi-step prediction with intermediate supervision)

Through controlled experiments across 27 BC benchmarks (Robomimic, Adroit, MetaWorld, etc.) with matched architectures, they show that C1 is not important, while C2 and C3 are the true drivers of GCP performance.
Based on this, they propose a Minimal Iterative Policy (MIP) that keeps only C2 + C3—achieving parity with flow models but with deterministic inference and far lower compute cost.
They also introduce manifold adherence as a diagnostic for how closely policy outputs stay near expert data manifolds.

**Strengths:**

Rigorous experimental isolation of C1, C2, C3 effects.

Unusually broad benchmark coverage (27 tasks).

Strong ablations: flow vs regression vs MIP.

Negative result clearly demonstrated and well-reasoned.

Practical takeaway: simple two-step deterministic policy matches flow performance.

“Manifold adherence” metric offers a novel lens on robustness.

**Weaknesses:**

All evaluations are offline imitation; no fine-tuning or real-robot rollouts to confirm closed-loop stability.

MIP’s hyperparameters (e.g. noise scale) are fixed; sensitivity analysis would help.

**Questions:**

Have you tried learning $\(t^*\)$ or using a schedule instead of a fixed value (0.9)?

Could MIP’s iterative refinement be stacked (3–4 steps) without re-introducing flow losses—does performance saturate after two steps?

---

> ### Author Response · Authors · 2025-11-19
> **Response to Reviewer Kdio**
>
> We would like to thank the reviewer heartily for the high quality review and valuable feedback.
> Here are our responses to the comments:
>
> ## Offline IL and Closed-Loop Stability
>
> > All evaluations are offline imitation. There is no RL fine-tuning or real-robot rollouts, which would be important to confirm closed-loop stability.
>
> Our main focus in this work is behavior cloning in standard offline IL benchmarks. All policies are evaluated in simulation in a fully closed-loop manner (for example, tasks like tool hang require very precise maneuvers).
>
> We agree that extending to RL fine-tuning and real-world control is important future work.
> However, they are beyond the scope of this paper which focuses on behavior cloning.
> Furthermore,
> In the revision, we will list other applications and other rationales for distributional learning in future work.
>
> ---
>
> ## MIP Hyperparameters and Sensitivity
>
> > MIP’s hyperparameters (e.g., noise scale) are fixed. A sensitivity analysis would help.
>
> We agree with this concern.
> In the current version, we simply pick a shared t* heuristically and didn't do much tuning since its performance is already close to flow.
> Here we report the ablation result on the choice of t* in MIP, where we tried both fixed t* and uniform sampling t* from [0.5, 0.9].
> MIP's performance is not very sensitive to the choice of t* and t*=0.9 appears to be a good choice.
>
> | Task      | $t^*=0.5$ | $t^*=0.6$ | $t^*=0.7$ | $t^*=0.8$ | $t^*=0.9$ | $t^* \sim \text{Uniform}(0.5, 0.9)$ |
> | --------- | --------- | --------- | --------- | --------- | --------- | ----------------------------------- |
> | Tool Hang | 0.65      | 0.59      | 0.66      | 0.70      | 0.77      | 0.67                                |
> | Transport | 0.50      | 0.55      | 0.65      | 0.60      | 0.75      | 0.63                                |
>
> Table: ablation result on the choice of $t^*$ in MIP. We report the average performance across 5 checkpoints with 3 random seeds on 3 architectures (DiT, Transformer, UNet). Tool hang and transport are picked from our benchmark since they demonstrate the highest performance gap between flow and regression. Task are state-based tasks with proficient human demonstrations.
>
> ---
>
> ## Learning or Scheduling $t^*$
>
> > Have you tried learning $t^*$ or using a schedule instead of a fixed value?
>
> We have not yet tried learning t* or using a schedule. We believe that the choice of t* is related to manifold adherence, and may be difficult to optimize purely from offline data without interaction with environment.
> Since with fixed t*, MIP already works well and match the performance of flow, we think it is okay to keep it fixed and make it a minimum tool to assist further analysis and research.
>
> ---
>
> ## More Iterative Steps for MIP
>
> > Can MIP’s iterative refinement be stacked (3–4 steps) without reintroducing flow losses? Does performance saturate after two steps?
>
> We also tried training MIP with 3 steps, using $t_{*,1} = 0.7$ and $t_{*,2} = 0.9$. Under our current computational budget, its performance is similar to the 2-step version and does not show significant gains.
> Theoretically, in the limit of infinitely many steps, MIP would approach a flow with zero initial noise. Given our setting, we do not see clear benefits beyond two steps and therefore keep MIP minimal for efficiency.
>
> | Task      | Flow (NFE=9) | MIP (NFE=3) | MIP (NFE=2) | Regression (NFE=1) |
> | --------- | ------------ | ----------- | ----------- | ------------------ |
> | Tool Hang | **0.80**     | 0.77        | 0.78        | 0.52               |
> | Transport | 0.87         | **0.90**    | 0.89        | 0.66               |
>
> Table: Averaged performance across 5 checkpoints with 3 random seeds on 3 architectures (DiT, Transformer, UNet). Tool Hang and Transport are state-based tasks with proficient human demonstrations. MIP (NFE=3) is trained with t1 = 0.7 and t2 = 0.9. MIP (NFE=2) is trained with t* = 0.9.

---

### Official Review · Reviewer_QhDz · 2025-10-31

**Soundness:** 2
**Presentation:** 3
**Contribution:** 2
**Rating:** 6
**Confidence:** 3

**Summary:**

This paper investigates why generative control policies often outperform regression-based ones in imitation learning. The authors aim to show (through controlled evaluations) that this advantage is not due to multimodal behaviors or higher expressiveness of generative models, but rather due to manifold adherence. They argue that, since control policies mainly map observations to actions, such complexity is unnecessary; what matters is incorporating some degree of stochasticity and supervised iterative computation. Based on this insight, they propose a minimal approach that achieves strong results on several continuous control benchmarks.

Overall, I find the contribution valuable for the community. The main evaluation, while not entirely conclusive, is well-designed and offers interesting (and somewhat counterintuitive) insights, showing that generative policies’ strength may not come from multimodality or expressiveness. The second contribution, the MIP framework, also seems practical and empirically validated. I have mixed opinions overall: some parts of the evaluation are not fully rigorous, but the results and reasoning are promising. I would give this paper an initial rating of 6 and look forward to further discussion with the authors and other reviewers.

**Strengths:**

1. I like the motivation of this work. Decomposing and analyzing the underlying reasons behind the success of generative policies is valuable, especially for RL and imitation learning, where understanding why a model works often matters more than just scaling it up. As often said, RL doesn’t necessarily benefit from ever-larger networks like LLMs or VLMs do  (IMO since the right actions or behaviors come from a subtle, structured distribution that doesn’t simply improve with more data or capacity..).

2. For the most part, the evaluation design is solid, and the conclusions drawn from the hypothesis testing seem meaningful and informative.

3. The proposed minimal algorithm looks sound and provides fresh insights into behavior cloning and how to simplify it (with stochasticity, and iterative computing but not necessarily model the whole distribution) without losing performance.

**Weaknesses:**

I list both my (tentative) weaknesses and questions together here, since some of them overlap, and a few points might just come from my not fully understanding certain parts of the evaluation.

1. On the conclusion that GCPs cannot really produce multi-modal actions, I have some concerns about the setup of evidence C. Even in deterministic environments and with deterministic policies, the underlying mapping from observation to action can still be stochastic or even multi-modal at the distribution level (e.g., multiple valid actions for the same state). So using this setting to claim the absence of multi-modality might be a bit too strong.

2. Related to that, the theoretical part on policy expressiveness seems to lean heavily on the assumption that there is no multi-modality (so that you can use the k-log-concave unimodal distribution arguments). But this then fully depends on the correctness of the earlier empirical conclusion about multi-modality. In practice, we do observe multimodal behaviors in pretrained GCPs, e.g. in the DP-like setups, so I’d like to see more clarification here.

3, For the claim that C2 supports C3 (i.e., noise injection can replace data augmentation), I’m not fully convinced. The noise here is injected on the action/control side or in the dynamics, while “data augmentation” in imitation learning can also mean visual, semantic, or context-level augmentation, which GCPs can implicitly provide. It might help to define more precisely what kind of “augmentation” you mean.

4. For the pixel-based control part where you show RCP can match GCP, I suspect this may partly come from the architectural alignment you used. If that’s the case, then H1 may not be as strong a takeaway as presented, it could just be an architecture effect. It would be good to clarify whether there is another implication here about the observation-to-action mapping.

5. To make the conclusions more universal, it would help to also test diffusion-based and transformer-based imitation policies. Even if they share the spirit of flow models, it would make the claim solid that the result is not specific to one generative family.

6. About the choice of $t_{*}=0.9$ is this environment-specific or a general setting? If it’s a trade-off hyperparameter, it would be good to say so and show a small sensitivity analysis

**Questions:**

Other than the points above, I also have a more general question. From the perspective of the true action generation process, it’s inherently a distribution. So, in principle, learning this distribution directly (as GCPs do) still seems like the optimal approach. The paper focuses on which components of the objective function work best, but from a first-principles standpoint, modeling the full action distribution still feels like the most appropriate formulation IMO

---

> ### Author Response · Authors · 2025-11-19
> **Response to Reviewer QhDz**
>
> We sincerely thank the reviewer for their constructive comments and dedication to the paper. We have addressed each point below:
>
> ## Evidence C and Multimodality
> We clarify that our goal is not to claim GCPs *cannot* be multimodal. Indeed, in long-horizon planning [1], generative models successfully learn multimodal behaviors.
> However, our results indicate that in standard control benchmarks used to claim GCP superiority (e.g., Diffusion Policy [5]), policies do not actually learn multimodal behaviors.
>
> * A: shows multimodality is not learned in standard pipelines.
> * B: shows explicitly degrading multimodality has negligible performance impact.
> * C: shows GCPs still outperform RCPs even when training data is forced to be unimodal.
>
> Together, this suggests that capturing multimodality is not the primary driver of GCP success in these settings. Instead, factors like **manifold-adherence bias** are responsible. Theorem 1 was motivated by Evidence C to explain why GCPs outperform RCPs even without multimodality.
>
> ## Theory and Unimodality Assumption
> We agree that real-world data can be multimodal. However, Theorem 1 is designed specifically to explain the phenomenon observed in Evidence C: why GCPs outperform RCPs even on *deterministic* datasets where multimodality is eliminated. The $k$-log-concave assumption is introduced to theoretically account for this specific gap.
>
> ## C2 as "Data Augmentation"
> We apologize for the confusion. We distinguish between environment-level data augmentation (e.g., image cropping) and augmentation of the **iterative generative process**.
> Previous work views generation as a dynamical system [2]. Our claim is that noise injection stabilizes *this* specific dynamics (mapping initial noise to final action) [3,4]. In the revision, we will include a figure clarifying that this "generative augmentation" is distinct from traditional task-level MDP data augmentation.
>
> ## Pixel-Based Control and Architecture Alignment
> We strongly agree. Once we align architectures (e.g., adding a shared ResNet encoder without specialized components), regression policies become highly competitive with GCPs.
> This is a core message of our paper: prior gaps in the literature were partly due to architectural mismatches (e.g., DiT/UNet vs. MLP/RNN) rather than pure parameterization differences. When architectures are aligned, RCPs often match GCPs.
>
> ## Additional Generative Baselines
> We have ensured our experiments cover standard backbone architectures (Transformer, DiT, UNet). We are happy to include more architectures up on reviewer's request.
> Regarding parameterization, we followed the reviewer's suggestion to evaluate SDE (DDPM) integrators, despite ODE (DDIM) being standard in control for efficiency.
> SDE provides a very minor performance improvement at the cost of ~50x more integration steps, which is generally impractical for real-time control.
>
> | Integrator | ODE, NFE=9 | SDE, NFE=500 |
> | :--------- | :--------- | :----------- |
> | Transport  | 0.77       | 0.78         |
> | Tool Hang  | 0.75       | 0.80         |
>
> **Table: Integrator Comparison.** Average performance (5 checkpoints, 3 seeds, 3 architectures) on the most challenging tasks. SDE requires 500 NFEs to saturate performance.
>
> ## Choice of $t^* = 0.9$
> The choice was heuristic, but our ablation study confirms that MIP is not highly sensitive to $t^*$, though $t^*=0.9$ performs best.
>
> | Task      | $t^*=0.5$ | $t^*=0.6$ | $t^*=0.7$ | $t^*=0.8$ | $t^*=0.9$ | $t^* \sim U(0.5, 0.9)$ |
> | :-------- | :-------- | :--- | :----| :----- | :--- | :------ |
> | Tool Hang | 0.65      | 0.59      | 0.66      | 0.70      | **0.77**  | 0.67                   |
> | Transport | 0.50      | 0.55      | 0.65      | 0.60      | **0.75**  | 0.63                   |
>
> **Table: Ablation on $t^*$.** Results averaged across 5 checkpoints, 3 seeds, and 3 architectures on state-based tasks with proficient human demonstrations.
>
> ## Full Action Distribution vs Our Focus
> We agree that distributional learning is conceptually appealing and beneficial for areas like RL exploration or large-scale VLA training [2].
> However, our empirical findings in single-task Behavior Cloning benchmarks (e.g., Push-T [5]) show that GCPs do not actually learn complex or multimodal distributions in practice.
> This observation (a) allows simpler algorithms like MIP to achieve competitive performance, and (b) highlights that other factors, such as manifold adherence, are the true drivers of GCP success in these regimes.
>
> [1] Janner et al., "Planning with Diffusion for Flexible Behavior Synthesis", ICML 2022.
>
> [2] Ren et al., "Diffusion Policy Policy Optimization", ICLR 2025.
>
> [3] Zhang et al., "Action Chunking and Exploratory Data Collection Yield Exponential Improvements in Behavior Cloning for Continuous Control", 2025.
>
> [4] Laskey et al., "DART: Noise Injection for Robust Imitation Learning", CoRL 2017.
>
> [5] Chi et al., "Diffusion Policy: Visuomotor Policy Learning via Action Diffusion", RSS 2023.

---

> > ### Comment · Reviewer_QhDz · 2025-11-26
> >
> > Thank you for the detailed response. My concerns have been addressed, and I’m happy to keep my positive rating.

---

### Official Review · Reviewer_ddUV · 2025-11-01

**Soundness:** 2
**Presentation:** 3
**Contribution:** 2
**Rating:** 4
**Confidence:** 4

**Summary:**

This paper seeks to understand the "secret sauce" behind the recent success of Generative Control Policies (GCPs) in robotics. While the field has largely attributed this success to the power of generative models (like diffusion) to handle multi-modality or complex representations, this paper systematically analyzes those common assumptions.

The authors argue that the generative, distributional aspect of these models is their least important feature. Instead, the performance leap comes from a powerful and previously under-appreciated combination of two other design choices: Supervised Iterative Computation (C3) and Stochasticity Injection (C2).

There're several claims in the paper

1. Analysis on multimodality of GCPs
The paper's argument is built in two parts. Firstly, the authors show that on tasks known to be multi-modal, GCPs show no significant performance advantage. Using a set of ablation experiments (e.g., showing that taking the mean action doesn't cause performance to collapse), they provide evidence that the models are not, in fact, learning distinct action modes.
The paper challenges the idea that the iterative nature of GCPs makes them deeper or more expressive. Through a theoretical argument (Theorem 1) and an empirical measurement of the Policy Lipschitz Constant, the authors show the exact opposite: successful GCPs learn smoother, less complex functions than their regression counterparts.

2. the authors introduce three core components of any GCP:
C1: Distributional Learning
C2: Stochasticity Injection (adding noise during training)
C3: Supervised Iterative Computation (refining an action over multiple steps)
The paper proposes simplified algorithm they call the Minimal Iterative Policy (MIP). MIP is designed to completely discard the complex generative component (C1) and rely only on C2 and C3.
The results show that this simple MIP matches the performance of the full, state-of-the-art Flow-based GCPs. This experiment conclusively demonstrates that the C2+C3 combination is the true engine of performance.

3. Finally, the paper provides a powerful explanation for why this C2+C3 combo is so effective.
They find that while MIP is no better than regression at simple imitation (i.e., they have the same validation loss), it is dramatically better at producing plausible actions when it encounters unfamiliar states. This iterative refinement process learns to fail gracefully, always steering its actions back toward the "manifold" (the space of valid expert actions). The iterative process (C3) fails on its own. The authors find that the injected noise (C2) is the critical ingredient that stabilizes the training, allowing the iterative refinement to learn its task without being derailed by compounding errors.

**Strengths:**

1. A key finding of this paper is that when using the same network architecture, traditional Regression Control Policies (RCPs) are highly competitive with modern Generative Control Policies (GCPs). This is an important contribution, as it helps correct a potential misconception in the field (stemming from earlier works that used different architectures for baselines) that diffusion policies inherently offer a large performance gain over standard behavior cloning on these tasks.

2. A major strength of this paper is its systematic methodology. The authors first propose a novel taxonomy to deconstruct GCPs into three key components, and then create 'ablation' models (MIP, RR, and SF) to empirically isolate and test the precise performance contribution of each component.

**Weaknesses:**

1. The paper's use of the Lipschitz constant as a metric for "expressivity" is a potential point of confusion. As you noted, the Lipschitz constant measures smoothness or robustness (how much the output can change for a small input change), not necessarily the representational capacity (the class of functions a model can learn). While their point (that GCPs find smoother solutions) is valid, their terminology can be debated.

2. The paper's theoretical argument, presented as Theorem 1 (Informal), feels underdeveloped. It attempts to formally decouple the number of iterative steps from an equivalent network's depth, but the premise of this connection to a diffusion policy's true expressiveness is not rigorously established. As presented, it functions more as a high-level intuition than a solid proof, making it a weak pillar for the paper's claims.

3. A significant limitation is the paper's narrow scope. The strong conclusion that Distributional Learning (C1) is the "least important factor" is derived entirely from a single-task, high-precision setting. This overlooks the primary strength of generative models: capturing complex, diverse distributions, which is fundamental for multi-task learning. The success of diffusion models in other domains (e.g., vision) is precisely their ability to model complex, multi-modal data distributions. It is highly plausible that C1 is, in fact, the critical component for scaling to multi-task behavioral distributions, making the paper's central finding a potential artifact of its limited, single-task evaluation.

4. A particularly counter-intuitive finding is the diffusion policy's failure to produce multi-modal behavior, given that diffusion models are expressly designed to capture complex data distributions. This observed uni-modal output may not be a failure of the model itself, but rather an artifact of the DDPM sampling process.
DDPM functions as a stochastic sampler (an SDE), and its Langevin dynamics term, while beneficial for error correction, could be the mechanism responsible for mode concentration. This stochastic term might be "correcting" the trajectory so effectively that it collapses different potential paths into a single, high-density outcome.
A crucial follow-up experiment would be to test the policy with a deterministic DDIM sampler (an ODE). This would remove the influence of the Langevin term, offering a clearer view of the true learned distribution. It's possible that this would reveal the multi-modality the model did learn, even if it simultaneously lowers task performance by removing the stochastic error correction.

Overall, this is an interesting paper with provocative findings. However, its conclusions feel brittle. They contradict the established behavior of diffusion models in other domains, a discrepancy likely stemming from the paper's narrow, single-task focus. The arguments lack full rigor, relying on conflated terminology (Lipschitz vs. expressivity) and informal theorems. The paper successfully isolates a powerful mechanism (C2+C3) for this specific setting, but it does not provide a generalizable understanding of generative policies in robotics.

**Questions:**

Besides the questions I raised in the weakness part.

Minor Point::
The paper's citation on line 159 for the concept of "straight interpolation" (or similar flow matching/rectified flow concepts) is incomplete. The authors cite only "Flow Matching" (Lipman et al. [3]), but this was one of several foundational works that introduced this concept concurrently.

[1]Albergo, Michael, and Eric Vanden-Eijnden. "Building Normalizing Flows with Stochastic Interpolants." ICLR 2023 Conference. 2023.
[2] Heitz, Eric, Laurent Belcour, and Thomas Chambon. "Iterative α-(de) blending: A minimalist deterministic diffusion model." ACM SIGGRAPH 2023 Conference Proceedings. 2023.
[3] Lipman, Yaron, et al. "Flow Matching for Generative Modeling." The Eleventh International Conference on Learning Representations.
[4] Liu, Xingchao, and Chengyue Gong. "Flow Straight and Fast: Learning to Generate and Transfer Data with Rectified Flow." The Eleventh International Conference on Learning Representations.

---

> ### Author Response · Authors · 2025-11-19
> **Response to Reviewer ddUV**
>
> We thank the reviewer for their constructive comments and detailed feedback. We have updated the manuscript to address the points raised regarding expressivity, citations, and multi-task settings.
>
> ## Lipschitz vs Expressivity
> We apologize for the confusion. In the revision, we clarify that high Lipschitzness is crucial for robotics as it dictates reactivity and precision in control. We have updated the terminology from "expressivity" to "**expressiveness of high Lipschitz behavior**." Theorem 1 demonstrates that, in an idealized setting, GCP parameterization does not inherently aid in expressing this essential feature compared to RCPs.
>
> ## Theorem 1 and Iterative Depth
> We have the formal theorem and proof in the appendix.
> Regarding expressivity, we emphasize that our theorem measures the **mapping** from the conditioning variable to the generated variable, rather than the complexity of the generated variable itself. This aligns with our empirical finding that policy multi-modality is often not learned even when the model allows for it.
>
> ## Scope of the C1 Conclusion (Multi-Task Generalization)
> While early claims about GCP superiority in control (e.g., Diffusion Policy [1], Flow Policy [3]) were drawn largely from single-task benchmarks like Push-T, we agree that validating these claims in modern multi-task regimes is critical.
>
> We evaluated fine-tuning the PI0 VLA on the **LIBERO benchmark** (130 tasks). As shown below, MIP achieves competitive performance with significantly fewer steps, and simple Regression performs surprisingly well. This reinforces our conclusion that capturing multi-modality is not the sole driver of performance, even in multi-task settings.
>
> | Task            | Libero Object | Libero Goal | Libero Spatial | Libero 10 |
> | :-------------- | :------------ | :---------- | :------------- | :-------- |
> | Regression      | 92.6          | 94.6        | 97.2           | 78.0      |
> | Flow (NFE=10)   | **97.4**      | 95.0        | 95.8           | 81.6      |
> | **MIP (NFE=2)** | 95.8          | **95.2**    | **97.6**       | **82.2**  |
>
> **Table: LIBERO Benchmark Results.** We implement MIP ($t^*=0.9$) and Flow using `lerobot`’s pi0 fine-tuning code (50 prediction steps, 5 action steps). Training runs for 50k gradient steps (8x H100). MIP matches Flow performance with 5x fewer inference steps.
>
> While GCPs may offer benefits in pre-training scalability, this evidence suggests that for downstream task success, the specific fitting of the conditional distribution is not the primary factor for improvement.
>
> ## Uni-Modality and DDPM vs ODE Sampling
> We utilized a deterministic integrator (ODE/DDIM) as it is standard practice in [1-3] for efficiency. Per your suggestion, we evaluated an SDE integrator (similar to DDPM) on our hardest tasks. We observed only minor improvements at the cost of massive inference overhead.
>
> | Integrator | ODE (NFE=9) | SDE (NFE=500) |
> | :--------- | :---------- | :------------ |
> | Transport  | 0.77        | 0.78          |
> | Tool Hang  | 0.75        | 0.80          |
>
> **Table: Integrator Comparison.** Average success on "Tool Hang" and "Transport" (3 seeds, 3 architectures). SDE requires ~50x more steps for marginal gain.
>
> **On Multi-modality:** In control, policies map high-dimensional observations to actions. With sparse data coverage, policies often achieve high success without learning the full multi-modal distribution. Furthermore, even "unimodal" policies produce diverse behaviors via state feedback loops. We validated this in the Kitchen environment:
>
> | Method                            | # Completion Orders |
> | :-------------------------------- | :------------------ |
> | Regression                        | 5                   |
> | MIP                               | 6                   |
> | Flow (Stochastic & Deterministic) | 6                   |
>
> **Table: Behavior Diversity.** Number of completion orders appearing >50 times in 1000 trials. All policies solve tasks with diverse orders.
>
> ## Citations
> We have added the missing references regarding straight interpolation and flow matching, including Albergo & Vanden-Eijnden, Heitz et al. (Iterative $\alpha$-(de)blending), Lipman et al. (Flow Matching), and Liu & Gong (Rectified Flow), acknowledging them as concurrent contributions.
>
> ### References
> [1] Chi et al., "Diffusion Policy: Visuomotor Policy Learning via Action Diffusion", RSS 2023.
>
> [2] Ze et al., "3D Diffusion Policy: Generalizable Visuomotor Policy Learning via Simple 3D Representations", RSS 2024.
>
> [3] Zhang et al., "FlowPolicy: Enabling Fast and Robust 3D Flow-Based Policy via Consistency Flow Matching for Robot Manipulation", AAAI 2025.

---

### Meta-Review · Area_Chair_uEgg · 2026-01-07

**Summary:**

Reviewers generally found the paper's systematic deconstruction of Generative Control Policies (GCPs) through controlled, architecture-matched experiments to be a valuable contribution. However, several key concerns initially hindered a consensus for acceptance:
- Reviewers `ddUV`, `zgJS`, and `QhDz` questioned if the finding that distributional learning (C1) is "least important" was an artifact of single-task, high-precision regimes.
- Concerns were raised regarding whether the observed lack of multimodality was inherent to the models or a result of specific sampling/inference choices, such as using ODE integrators over SDE.
- Reviewer `ddUV` noted issues with terminology (Lipschitz constant vs. expressivity) and felt the informal theorem regarding iterative depth lacked sufficient rigor.
- Reviewers `Kdio` and `zgJS` highlighted the absence of real-robot rollouts, RL fine-tuning, or long-horizon planning to confirm closed-loop stability and generality.

Prior to the rebuttal, scores were split between Marginally Below (4) and Accept (8/6). In the rebuttal, the authors effectively addressed these points by adding a multi-task LIBERO evaluation, reporting SDE vs. ODE integrator results, clarifying their theoretical framing of high-Lipschitz behavior, and providing robustness and behavior diversity metrics.

The AC has carefully reviewed the paper, prior rebuttal reviews, rebuttals, and discussion, and agrees that the rebuttals substantially address the technical concerns, leading to a final recommendation of Accept.

**Reviewer Concerns:**

Concerns centered on (i) whether the “C1 is least important” conclusion is too strong or too tied to limited regimes (`ddUV`, `zgJS`, `QhDz`), (ii) multimodality evidence and the role of sampling/inference (`ddUV`, `QhDz`, `zgJS`), (iii) theory/terminology rigor (`ddUV`, `QhDz`), and (iv) practical scope (offline IL only; sensitivity; robustness; RL fine-tuning/long-horizon) (`Kdio`, `zgJS`). The rebuttal addresses most of these with clearer framing around multimodality, stronger ablations (integrator ODE vs SDE; behavior diversity), added sensitivity and multi-step MIP results, citation fixes, and a multi-task LIBERO experiment. That said, it is worth noting that parts of the community increasingly view LIBERO performance as somewhat saturated and potentially subject to benchmark-specific overfitting, so while the LIBERO results help broaden scope, stronger evidence on newer/less-saturated multi-task settings (e.g., SimplerEnv, RoboTwin) and/or real-world multi-task evaluations would further strengthen the paper’s generality claims.

**Reviewer Scores:**

Given the rebuttal and discussion, I expect `zgJS` stays at 8, `Kdio` stays at 6, and `QhDz` stays at 6. `ddUV` is likely to move from 4 to around 5–6 due to the added multi-task evidence, integrator checks, clarified terminology, and expanded ablations, though they may still view some central claims as potentially regime-dependent.

---

### Decision · Program_Chairs · 2026-01-26

Accept (Poster)